# Regularized Training of Intermediate Layers for Generative Models for Inverse Problems

**Sean Gunn**                                                                  *gunn.s@northeastern.edu*
*Khoury College of Computer Sciences*
*Northeastern University*

**Jorio Cocola**                                                                  *jcocola@seas.harvard.edu*
*Institute for Applied Computational Science*
*Harvard University*

**Paul Hand**                                                                  *p.hand@northeastern.edu*
*Department of Mathematics and Khoury College of Computer Sciences*
*Northeastern University*

**Reviewed on OpenReview:** *https://openreview.net/forum?id=cKsKXR28cG*

## Abstract

Generative Adversarial Networks (GANs) have been shown to be powerful and flexible priors when solving inverse problems. One challenge of using them is overcoming representation error, the fundamental limitation of the network in representing any particular signal. Recently, multiple proposed inversion algorithms reduce representation error by optimizing over intermediate layer representations. These methods are typically applied to generative models that were trained agnostic of the downstream inversion algorithm. In our work, we introduce a principle that if a generative model is intended for inversion using an algorithm based on the optimization of intermediate layers, it should be trained in a way that regularizes those intermediate layers. We instantiate this principle for two notable recent inversion algorithms: Intermediate Layer Optimization and the Multi-Code GAN prior. For both of these inversion algorithms, we introduce a new regularized GAN training algorithm and demonstrate that the learned generative model results in lower reconstruction errors across a wide range of under-sampling ratios when solving compressed sensing, inpainting, and super-resolution problems.

## 1 Introduction

The task in an inverse problem is to estimate an unknown signal given a (possibly noisy) set of measurements of that signal. In practice, inverse problems are often ill-posed, which often require the incorporation of prior information about the target signal to recover a reasonable estimate of it.

Deep generative models have demonstrated remarkable performance when used as priors for solving inverse problems (Bora et al., 2017; Athar et al., 2018; Hand et al., 2018; Menon et al., 2020; Mardani et al., 2018; Mosser et al., 2020; Pan et al., 2021). Typically, a generative modeling-based approach for inverse problems has two phases: a training phase and an inversion/deployment phase. In the first phase, a generative network is trained on a dataset of images. After training, the network parameters are typically fixed and an optimization problem involving the known forward model is solved to estimate an unknown signal of interest. An advantage of this class of methods is that the prior can be trained entirely in an unsupervised manner and without previous knowledge of the specific inverse problem that needs to be solved downstream. This allow the trained network to be used for a variety of inverse problems.

Some generative networks explicitly map a low-dimensional input to a high-dimensional signal space. The outputs of the generative network, therefore, have a low intrinsic dimensionality. While this is useful for regularizing the inverse problem, it can limit the expressivity of the network and lead to *representation error*, the error between a target signal and the closest signal in the range of the generative network.

Several recent methods have attempted to reduce representation error by introducing optimization algorithms that enlarge the search space of the inversion algorithm. These include methods that optimize both over the input of the network and over the activations of one or more intermediate layers. Examples of this class of inversion methods include Intermediate Layer Optimization (ILO) (Daras et al., 2021), GAN Surgery (Smedemark-Margulies et al., 2021), and the Multi-code GAN Prior (mGANprior) (Gu et al., 2020). These algorithms were applied to state-of-the-art off-the-shelf GANs, such as PGGAN and StyleGAN (Karras et al., 2018; 2020), which were trained agnostic to the specific optimization algorithm that would be used for inversion. Consequently, the inversion algorithms optimize over a region of the space of intermediate presentations that were not directly regularized during training. This suggests that it may be possible to improve the reconstruction performance of these algorithms by explicitly regularizing the intermediate layers during the training phase. In this paper, we confirm this hypothesis.

We consider the problem of training a GAN that will be used for solving inverse problems via an inversion algorithm that optimizes over intermediate layers. We put forward the following principle termed *Regularized Training of Intermediate Layers (RTIL)*:

*If a GAN's intermediate layers are optimized during inversion, they should be regularized during training.*

We use this principle to derive new training algorithms for GANs that are intended for solving inverse problems. For ILO and mGANprior, we introduce a new training algorithm for StyleGAN2 and PGGAN, respectively. For these new trained GANs, we achieve improved reconstruction quality relative to the corresponding GANs trained without the principle. We note that this paper is not about regularizing GANs in order to stabilize training and aiding the convergence of gradient descent (as in Gulrajani et al. (2017a); Mescheder

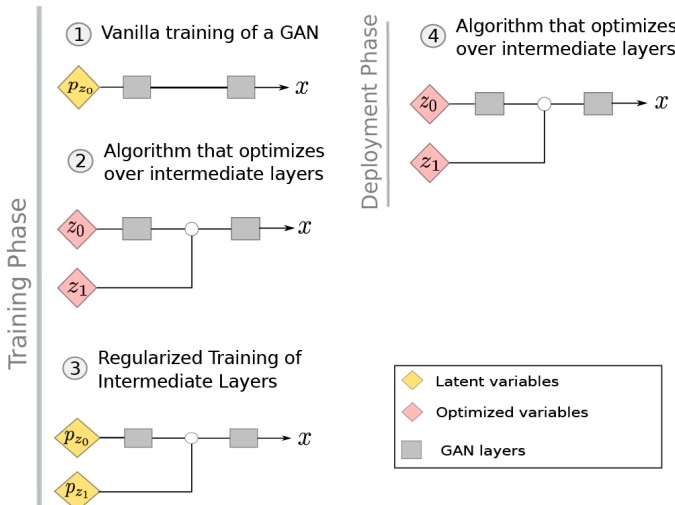

Figure 1: The workflow for using Regularized Training of Intermediate Layers (RTIL) to develop new training algorithms of GANs that will be used for inversion. See the text for details.

et al. (2018)). Rather, those regularization methods can be used in conjunction with RTIL to improve upon approximate learned inference for a specific algorithm used in solving inverse problems.

Our proposed principle may lead to the following workflow for solving inverse problems with GANs. First, train a generative network with latent variable $z_0$, sampled from a latent distribution $p_{z_0}$, and outputting signals $x$ from the target distribution (Figure 1 Step 1). Second, explore various inversion algorithms, including some that optimize over intermediate layers by introducing an additional optimization variable $z_1$ (Figure 1 Step 2). Algorithms of this type include ILO and mGANprior. Third, if such an algorithm provides competitive performance for inversion, then use RTIL to devise a new GAN training algorithm. This can be achieved by introducing a new latent variable $z_1 \sim p_{z_1}$ where $p_{z_1}$ is an appropriate probability distribution (Figure 1 Step 3). Finally, during deployment, use this trained generative network for inverse problems via the selected inversion algorithm (Figure 1 Step 4).

This workflow demonstrates a way to use recently introduced inversion algorithms in order to inspire new training algorithms for GANs. It provides additional ways of training GANs knowing that they will be used

for inversion, and it provides a way to enable some empirically successful inversion algorithms are operating in a more principled manner by ensuring they are searching over a space of parameters that have been suitably regularized.

The contributions of this paper are the following.

- We introduce Regularized Training of Intermediate Layers, a principle for training deep generative networks that are intended to be used for inverse problems when solved by an algorithm that optimizes over intermediate layer representations.

- In the case of Intermediate Layer Optimization, we use our principle to devise a novel GAN training algorithm. With the resulting trained GAN, we demonstrate lower reconstruction errors (compared to GAN training without the principle) for compressed sensing, inpainting, and super resolution over a wide range of under sampling ratio.

- We show the versatility of the method by repeating the same contribution in the case of the Multi-Code GAN Prior.

- We illustrate the benefits of compressed sensing with RTIL by theoretically showing that a model trained without regularizing intermediate layers has a strictly larger reconstruction error than a model where the intermediate layers were regularized during training. For simplicity, this result is established in the case of a two layer linear neural network trained in a supervised setting.

## 1 .1 Related Works

In recent years, various approaches have been proposed for dealing with the representation error when solving inverse problems with a learned generative prior. As described above, one class of approaches focuses on enlarging the latent space dimension, either at training or at inversion time (Athar et al., 2018; Dhar et al., 2018; Hussein et al., 2020). On the other hand, another notable class of approaches recently put forward is based on flow-based invertible neural networks. These generative networks have invertible architectures and a latent space with the same size of the image space, and thus have zero representation error (Ardizzone et al., 2019; Asim et al., 2020; Ma & Le Dimet, 2009; Kelkar et al., 2021; Shamshad et al., 2019; Whang et al., 2021a;b; Helminger et al., 2021). Both these type of approaches attempt to limit the representation error during training or inversion.

Similarly to flow-based models, score networks and subsequent variants (Song & Ermon, 2019; 2020; Song et al., 2020) have support on the entire signal space, allow for conditional sampling and likelihood estimation, and have been object of recent interests for their use in inverse problems (Ramzi et al., 2020; Jalal et al., 2021a;b).

The increased representation power of the above-mentioned generative networks, usually comes at a price of increased computational cost, both during the training and the inversion phase. In contrast, our proposed principle leads to training algorithms that have essentially the same computational cost as the standard ones, also leaving the cost of the inversion algorithms the same.

## 2 Generative Models and Optimization Algorithms for Inverse Problems

### 2 .1 Background

We consider the problem of training a GAN for the use of a prior for solving inverse problems. We consider a GAN $G$ which maps a latent space $\mathbb{R}^{n_0}$ to an image space $\mathbb{R}^{n_d}$, where $n_0 \ll n_d$. In this paper, we focus on the linear imaging inverse problems of compressed sensing, inpainting, and super resolution, though our proposed method also applies to nonlinear inverse problems such as phase retrieval, and inversion problems about non-image signals. We consider the general linear inverse problem of recovering an image $\boldsymbol{x} \in \mathbb{R}^{n_d}$ from a set of linear measurements $\boldsymbol{y} \in \mathbb{R}^m$, given by $\boldsymbol{y} = \mathcal{A}(\boldsymbol{x})$, where $\mathcal{A} : \mathbb{R}^{n_d} \to \mathbb{R}^m$ is a forward linear measurement operator. We study only the noiseless case, but our method easily extends to the case with measurement noise. In this paper, we study measurement operators $\mathcal{A}$ of the following form:

- Compressed Sensing: $\mathcal{A} = \mathbf{A} \in \mathbb{R}^{m \times n_d}$, $\mathbf{A}$ is a random matrix that samples from a known distribution with $m < n_d$.

- Inpainting: $\mathcal{A} = \boldsymbol{M} \in \mathbb{R}^{m \times n_d}$, $\boldsymbol{M}$ is a masking matrix with binary entries.

- Super-Resolution: $\mathcal{A} = \boldsymbol{S}_{m_\downarrow} \in \mathbb{R}^{m \times n_d}$ where $\boldsymbol{S}_{m_\downarrow}$ is the downsampling operator with downsampling factor $m_\downarrow$.

An estimate of $x$ can be recovered by finding the image in the range of $G$ that is most consistent with the measurements $\boldsymbol{y}$ in the following sense, as introduced in (Bora et al., 2017). First, solve

$$\hat{\boldsymbol{z}}_0 = \arg\min_{\boldsymbol{z}_0} \ \|\boldsymbol{y} - \mathcal{A}(G(\boldsymbol{z}_0))\|, \tag{1}$$

then, the estimate of $\boldsymbol{x}_0$ is given by $G(\boldsymbol{z}_0)$.

As mentioned in the introduction, a difficulty of this optimization approach is that the estimated images are constrained to live within the range of $G$, which is a $n_0$-dimensional manifold in $\mathbb{R}^{n_d}$. Most images $x_0$ will not live exactly in this range, and thus the method is limited by the representation error $\min_{\boldsymbol{z}_0} \|\boldsymbol{x}_0 - G(\boldsymbol{z}_0)\|$. In the next sections, we review two recent algorithms for mitigating representation error during inversion. For ease of exposition, we will discuss the case where only one intermediate layer representation is optimized. We write $G = g_1 \circ g_0$ where $g_0 : \mathbb{R}^{n_0} \to \mathbb{R}^{n_1}$ and $g_1 : \mathbb{R}^{n_1} \to \mathbb{R}^{n_d}$.

## 2.2 Intermediate Layer Optimization (ILO)

Given a trained generative model $G = g_1 \circ g_0$, Intermediate Layer Optimization (ILO) extends the range of the generative model by sequentially optimizing over each layer of the network, as demonstrated in Algorithm 1. The initial step begins exactly as in (Bora et al., 2017) by optimizing over the input vector $\boldsymbol{z}_0 \in \mathbb{R}^{n_0}$. The solution is obtained in line 3, by initializing a $\boldsymbol{z}_0 \sim \mathcal{N}(0, \boldsymbol{I}_{n_0})$, then optimizing the loss with gradient descent. After the solution $\hat{\boldsymbol{z}}_0$ is obtained, the algorithm searches for a perturbation $\hat{\boldsymbol{z}}_1$ of $g_0(\hat{\boldsymbol{z}}_0)$ that further minimizes the reconstruction error (line 4). The final approximation of the target image $\boldsymbol{x}$ is given by $g_1(\hat{\boldsymbol{z}}_1 + g_0(\hat{\boldsymbol{z}}_0))$. As the authors point out, there are multiple ways where ILO can be regularized, including by an L1 penalty in the intermediate representation, or via early stopping. For the present paper, we use early stopping, as this was the method used by the publicly available code from the authors. Throughout this paper, we use the code provided by the authors when solving ILO.

## 2.3 Multi-Code GAN (mGanPrior)

Multi-Code GAN Prior is an inversion method that simultaneously optimizes over multiple latent codes and composes their corresponding intermediate feature maps with adaptive channel importance. This effectively extends the expressivity of the network by giving it a higher dimensional input space and additional parameters to control the importance of each channels in the intermediate layer representation.

Assume we are given a pre-trained generative model $G = g_1 \circ g_0$, where the output of the first layer $g_0$ has dimension $H_1 \times W_1 \times C_1$ with $C_1$ being the number of channels. Furthermore, chose $N$ latent codes $\{\boldsymbol{z}_0^k\}_{i=1}^N \in \mathbb{R}^{n_0}$ and channel importance $\{\boldsymbol{\alpha}^k\}_{i=1}^N \in \mathbb{R}^{C_1}$. Then the Multi-Code GAN Prior extended architecture computes $g_1(\sum_{k=1}^N g_0(\boldsymbol{z}_0^k) \odot \boldsymbol{\alpha}^k)$ where $\{g_0(\boldsymbol{z}_0^k) \odot \boldsymbol{\alpha}^k\}_{ijc} = \{g_0(\boldsymbol{z}_0^k)\}_{ijc} \cdot \{\boldsymbol{\alpha}^k\}_c$ is channel-wise multiplication, $i, j$ are spatial location, and $c$ is the channel index.

---

**Algorithm 1** Intermediate Layer Optimization (ILO) for Compressed Sensing (Daras et al., 2021).

---

1: **Input:** $G = g_1 \circ g_0$, measurement matrix $\mathbf{A} \in \mathbb{R}^{m \times n_d}$, compressed measurements $\boldsymbol{y}$.

2: **Output:** estimated image $\hat{\boldsymbol{x}}$

3: $\hat{\boldsymbol{z}}_0 = \arg\min\limits_{\boldsymbol{z}_0} \ \|\boldsymbol{y} - \mathbf{A}g_1(g_0(\boldsymbol{z}_0))\|$ *Initialize at* $\boldsymbol{z}_0 \sim \mathcal{N}(0, \boldsymbol{I}_{n_0})$

4: $\hat{\boldsymbol{z}}_1 = \arg\min\limits_{\boldsymbol{z}_1} \ \|\boldsymbol{y} - \mathbf{A}g_1(\boldsymbol{z}_1 + g_0(\hat{\boldsymbol{z}}_0))\|$ *Initialize at* $\boldsymbol{z}_1 \sim \mathcal{N}(0, \boldsymbol{I}_{n_1})$

5: Return: $\hat{\boldsymbol{x}} = g_1(\hat{\boldsymbol{z}}_1 + g_0(\hat{\boldsymbol{z}}_0))$

---

During the inversion phase, the Multi-Code GAN Prior method optimizes over both the latent vectors $\{\boldsymbol{z}_0^k\}_{i=1}^N$ and the channel importance $\{\boldsymbol{\alpha}^k\}_{i=1}^N$ (Algorithm 2).

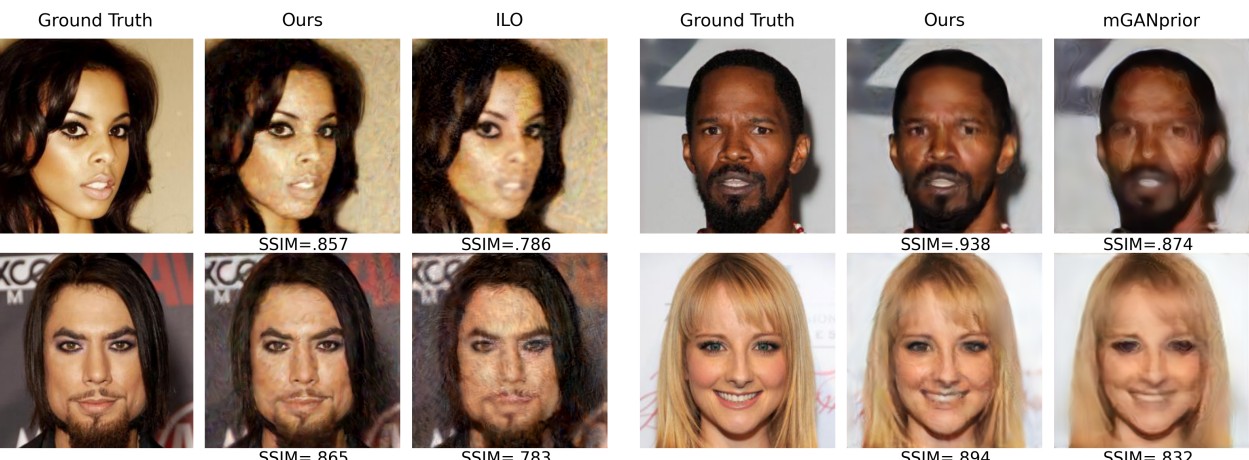

(a) Comparison between ILO-RTIL (ours) and ILO for compressed sensing for 3% of measurements.

(b) Comparison between mGANprior-RTIL (ours) and mGANprior for compressed sensing for 5% of measurements.

Figure 2: Results on the left compare ILO-RTIL to ILO and then on the right mGANprior-RTIL to mGANprior.

## 3 RTIL

In this section, we present how the principle of Regularized Training of Intermediate Layers (RTIL) inspires training algorithms for GANs that are intended for inversion by Intermediate Layer Optimization (Section 3 .1) and mGAN-prior (Section 3 .2).

We consider the case of a practitioner having chosen, after some initial exploration, a base generative network $G$ for use as prior in solving inverse problems. For simplicity, in this section we will consider $G = g_1 \circ g_0$ where $g_0 : \mathbb{R}^{n_0} \to \mathbb{R}^{n_1}$ and $g_1 : \mathbb{R}^{n_1} \to \mathbb{R}^{n_d}$. The input latent vectors $z_0$ of the network $G$ are sampled from $p_{z_0}$ (e.g. $p_{z_0} = \mathcal{N}(0, I_{n_0})$) and $\theta$ is the set of trained parameters.

**Algorithm 2** Multi-Code GAN (mGANprior) Gu et al. (2020).

1: **Input:** Trained network $G = g_1 \circ g_0$, latent codes $\{z_0^k\}_{k=1}^N \in \mathbb{R}^{n_0}$ , $\{\alpha^k\}_{k=1}^N \in \mathbb{R}^{n_1}$, measurement matrix $\mathbf{A} \in \mathbb{R}^{m \times n_d}$, compressed measurements $\boldsymbol{y}$.

2: **Output:** estimated image $\hat{\boldsymbol{x}}$

3: *Initialize* $\{z_0^k\}_{k=1}^N \sim p_{\boldsymbol{z}}, \{\alpha^k\}_{k=1}^N \sim p_{\boldsymbol{\alpha}}$

4: $\hat{\boldsymbol{z}}, \hat{\boldsymbol{\alpha}} = \underset{\{z_0^k\}_{k=1}^N, \{\alpha^k\}_{k=1}^n}{\arg\min} \|\boldsymbol{y} - \mathbf{A} g_1(\sum_{k=1}^n g_0(z_0^k) \odot \alpha^k)\|$

5: Return: $\hat{\boldsymbol{x}} = g_1(\sum_{k=1}^N g_0(\hat{z}_0^k) \odot \hat{\boldsymbol{\alpha}}^k)$

We assume, furthermore, that the practitioner has selected an inversion algorithm that optimizes over the latent variable $z_0$ and the intermediate layer between $g_0$ and $g_1$ of $G$. Optimizing over the intermediate layer corresponds to introducing a free variable $z_1$ between $g_0$ and $g_1$ (Figure 3).

The RTIL principle states that if one intends to solve inverse problems by means of intermediate layer optimization algorithms, then *intermediate layers optimized over during inversion should be regularized during training.*

This principle can be used to design a new training algorithm in the following manner. We identify the additional free variable, $z_1$, used for optimization over the intermediate layer, consider it as a latent variable of the generative model, and provide it with a simple distribution $p_{z_1}$. For example, this could result in a generative model $\widetilde{G} : \mathbb{R}^{n_0} \times \mathbb{R}^{n_1} \to \mathbb{R}^{n_d}$, such that $\widetilde{G}(z_0, z_1) = g_1(z_1 + g_0(z_0))$, but could have alternative functional forms. This generative model reduces to the base generative model $G : \mathbb{R}^{n_0} \to \mathbb{R}^{n_d}$ if $z_1$ is suitably chosen, for example if $z_1 = 0$.

The introduction of the latent variable $z_1$ explicitly increases the dimensionality of the latent space. In practice, training latent variable models with high-dimensional latent spaces can be challenging and require careful regularization (Athar et al., 2018). We address this difficulty by concurrently training the lower and higher dimensional models $G_\theta$ and $\widetilde{G}_\theta$, which share trainable weights $\theta$ (Figure 3).

We train $G_\theta$ and $\widetilde{G}_\theta$ via the following minimax formulation (Goodfellow et al., 2014)

$$\min_\theta \max_\Theta \mathbb{E}_{\boldsymbol{x} \sim p_{\boldsymbol{x}}} \big[ \log D_\Theta(\boldsymbol{x}) \big] + \mathbb{E}_{\substack{z_0 \sim p_{z_0} \\ z_1 \sim p_{z_1}}} \big[ \lambda \log(1 - D_\Theta(G_\theta(z_0))) + (1 - \lambda) \log(1 - D_\Theta(\widetilde{G}_\theta(z_0, z_1))) \big], \quad (2)$$

where $D_\Theta : \mathbb{R}^{n_d} \to \mathbb{R}$ is the discriminative network and $\lambda \in [0, 1]$ is a hyperparameter. This method could be extended to alternative GAN and non-GAN formulations (Arjovsky et al., 2017; Gulrajani et al., 2017b; Bojanowski et al., 2018).

Once $\widetilde{G}$ is trained, a practitioner could solve an inverse problem with forward operator $\mathcal{A}$ by solving

$$\hat{z}_0, \hat{z}_1 = \arg\min_{z_0, z_1} \ \|\boldsymbol{y}_0 - \mathcal{A}(\widetilde{G}(z_0, z_1))\|,$$

using the selected optimization algorithm, resulting in $\widetilde{G}(\hat{z}_0, \hat{z}_1)$ as the estimate of the signal.

In this section, we considered the case where only one intermediate layer was optimized. The proposed method can directly extend to the case where multiple layers are optimized.

We will next describe the details of the application of RTIL in the case of Intermediate Layer Optimization and the Multi-Code GAN Prior.

### 3.1 RTIL for ILO

In the case of Intermediate Layer Optimization (ILO), we now present how to use RTIL to design a GAN training algorithm. Consider the base generative network $G(z_0) = g_1(g_0(z_0))$, to be used for inversion with Intermediate Layer Optimization. ILO (Algorithm 1) extends the range of the network by optimizing $g_1(z_1 + g_0(z_0))$ over latent variables $z_0$ and $z_1$. Consequently, we consider the higher dimensional generative model

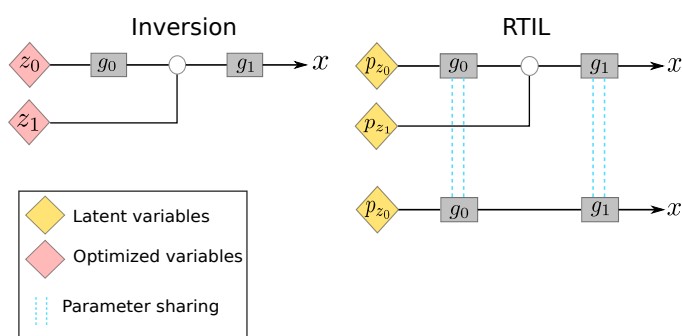

Figure 3: Visual representation of the RTIL principle. An inversion method that optimizes over intermediate layers informs the training of a family of generative networks adapted to ILO inversion method.

$$\widetilde{G}(z_0, z_1) = g_1(z_1 + g_0(z_0)), \ z_0 \sim \mathcal{N}(0, \boldsymbol{I}_{n_0}), \ z_1 \sim \mathcal{N}(0, \sigma^2 \boldsymbol{I}_{n_1}),$$

where $\sigma^2$ is a hyperparameter. Then simultaneously the lower dimensional model $G(z_0)$ and higher dimensional model $\widetilde{G}(z_0, z_1)$ are trained with the min max formulation above. Note $z_0 \sim p_{z_0}$ and $z_1 \sim p_{z_1}$ where Figure 3 depicts this process.

### 3.2 RTIL for mGANprior

In the case of Multi-Code GAN Prior method, we now present how to use RTIL to design a GAN training algorithm. Consider the base generative network $G$ to be used for inversion with the Multi-Code GAN Prior method. The mGANprior Algorithm (Algorithm 2) extends the range of the network by optimizing $g_1(\sum_{k=1}^{N} g_0(z_0^k) \odot \boldsymbol{\alpha}^k)$ over latent variables $z_0$ and $z_1$. Consequently, training the higher dimensional model

yields

$$\widetilde{G}(\boldsymbol{z}_0^1, \ldots, \boldsymbol{z}_0^N, \boldsymbol{\alpha}^1, \ldots, \boldsymbol{\alpha}^N) = g_1(\sum_{k=1}^{N} g_0(\boldsymbol{z}_0^k) \odot \boldsymbol{\alpha}^k), \ \boldsymbol{z}_0^k \sim \mathcal{N}(0, \boldsymbol{I}_{n_0}), \ p_{\boldsymbol{\alpha}'} \sim \mathrm{Dir}_N(1).$$

where each vector $\boldsymbol{\alpha}^k$ is taken to be $\boldsymbol{\alpha}^k = \{\boldsymbol{\alpha}'\}_k \cdot \mathbf{1}$ where $\mathbf{1}$ is the vector of all ones and $\{\boldsymbol{\alpha}'\}_k$ is the $k$-th entry of $\boldsymbol{\alpha}' \in \mathbb{R}^N$. The vector $\boldsymbol{\alpha}'$ is sampled from $p_{\boldsymbol{\alpha}'} \sim \mathrm{Dir}_N(1)$, where $\mathrm{Dir}_N(1)$ is the flat Dirichlet distribution, i.e. the uniform distribution over the $(N-1)$-dimensional simplex. Note this leads to each channel being weighted equally during training. The training process is depicted in the appendix, Figure 13.

## 4    Experiments

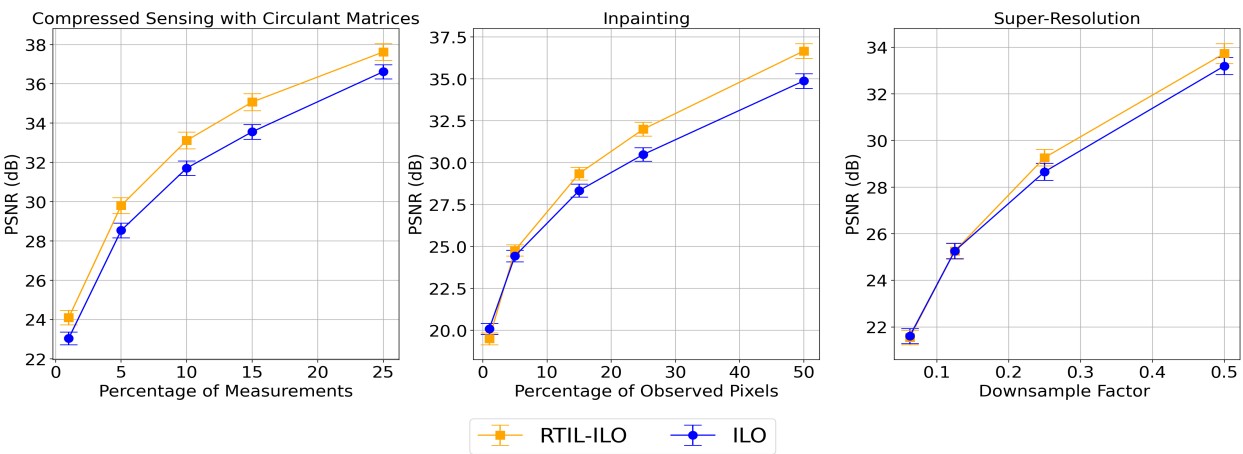

Figure 4: Performance of ILO-RTIL and vanilla trained ILO for Compressed sensing, inpainting, and super resolution for various under-sampling ratios. ILO-RTIL increases performances or ties for each under-sampling ratio with respect to PSNR across each of the inverse problems compared to ILO. The vertical bars indicate 95% confidence intervals.

We observe that Regularized Training of Intermediate Layers is not tied to any specific architecture or training procedure, and while, until this point, we have only presented it on two-layer networks for easiness of exposition, in this section we demonstrate its successful application on two state-of-the-art architectures on different imaging recovery problems. Specifically, we conduct extensive experiments comparing RTIL versus vanilla training for compressed sensing, inpainting, and super-resolution, and for two different inverse methods Multi-Code GAN Prior (Section 2 .3) and Intermediate Layer Optimization (ILO) (Section 2 .2) to demonstrate the effectiveness of our method. The generative model architecture used for the mGANprior is PGGAN (Karras et al., 2018) and for ILO is styleGAN-2 (Karras et al., 2020). All models were trained on FFHQ data set (Karras et al., 2019) and tested on CelebA-HQ dataset (Karras et al., 2018), at an image size of 256x256x3. The choice of architectures were based on the experimental section of the original ILO (Daras et al., 2021) and mGANprior (Gu et al., 2020) papers. Refer to the appendix for the architecture details for training networks using RTIL, as well as hyperparameters chosen for inversion methods.

For all experiments, compressed sensing use partial circulant measurement matrices with random signs Daras et al. (2021), inpainting the pixels are missing at random, and downsample factor corresponds to how much the height and width of the original image was reduced to. All experiments were trained with $\lambda = \frac{1}{2}$ equation 2. The code is hosted on a GitHub repository https://github.com/g33sean/RTIL .

### 4 .1    Results ILO-RTIL

Results in this section correspond to Figure 4, where the results are averaged over 100 images randomly sampled from CelebA-HQ. Note, the hyperparameter was set $\sigma^2 = 1$ during training for all experiments.

Compressed Sensing - ILO-RTIL demonstrates an increase in reconstruction performance across each under-sampling regime. The largest increase in performance with respect to PSNR occur at 25% (1.28 dB), 15% (1.59 dB), and 10% (1.3 dB's) measurements. For qualitative results, please refer to Figure 2(a). Inpainting-ILO-RTIL demonstrates in an increase in reconstruction performance with 4 out of the 5 sampling regimes. There is an increase in PSNR at 50% (1.88 dB), 25% (1.59 dB's), 15% (.94 dB's), and 5% (.29 db) of observed pixels. At 1% of observed pixels, there is a decrease in performance in reconstruction of .5 dB's, but the error bars overlap each other, indicating there is no significant improvement. For qualitative results, please refer to Figure 16. Super-Resolution - ILO-RTIL shows a slight increase in performance over the entire measurement regime compared to ILO, the most significant occurs at $\frac{1}{4}$ downsampling factor where on average the increase in reconstruction is .77 dB's and the 95% error bars are separated. All other sampling regime achieves comparable performance between ILO-RTIL and ILO. For qualitative results, please refer to the appendix Figure 18.

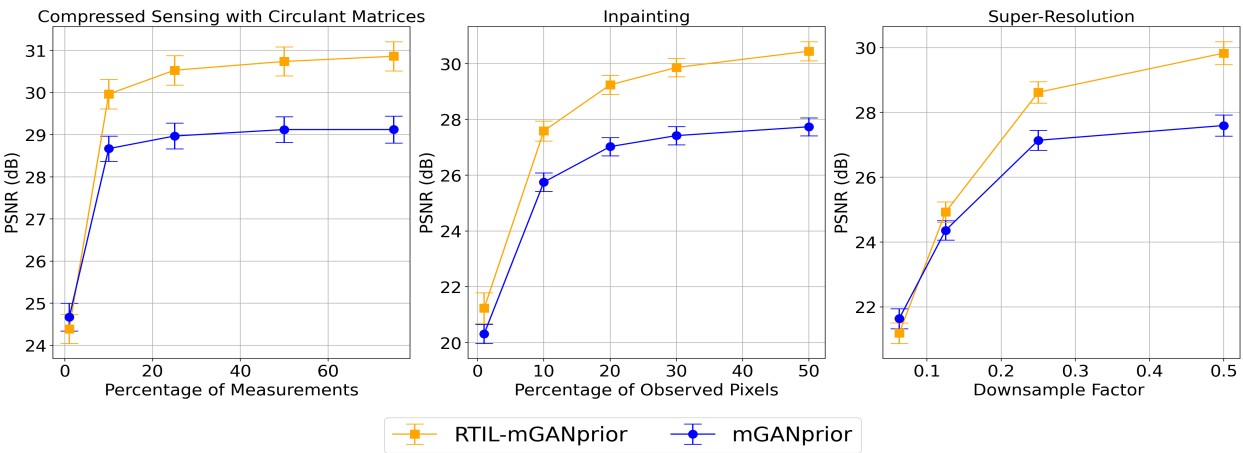

Figure 5: Performance of mGANprior-RTIL and vanilla trained mGANprior for compressed sensing, inpainting, and super resolution for various under-sampling ratios. mGANprior-RTIL increases performances over vanilla mGANprior with respect to PSNR over each under-sampling ratio, except for super-resolution problems at low under-sampling ratio's. The vertical bars indicate 95% confidence intervals.

## 4.2 Results RTIL-Multi-Code

Results in this section correspond to Figure 15, where the results are average over 100 images randomly sampled from CelebA-HQ and all experiments use $N = 20$ latent codes. Compressed Sensing - mGANprior-RTIL has a significant improvement in reconstruction performance over mGANprior. For each under-sampling ratio, our method increases performance by at least 1.6 dB's, which clearly separates the error bars. On the other hand, at 1% of measurements where both methods achieve the similar performance, which is seen by overlapping error bars. For qualitative results, refer to Figure 2(b). Inpainting- mGANprior-RTIL demonstrates a significant improvement in reconstructing over mGANprior clearly being able the error bars. Our method yields an increase in performance with respect to PSNR of 2.5 dB's at 50%, 2.23 dB's 30%, 1.99 dB's at 20%, and 1.58 dB's at 10% observed pixels. For 1% of observed pixels, these both methods yield similar results. Lastly, qualitative results can be seen on Figure 17. Super-Resolution- mGANprior-RTIL show substantial improvement with image reconstruction at downsampling factor of $\frac{1}{2}$ and $\frac{1}{4}$, increase of 2.25 dB's and 1.51 dB's respectively. However, at $\frac{1}{16}$ down-sampling factor mGANprior outperforms mGANprior-RTIL slightly, but within the error bars. For qualitative results, please refer to the appendix Figure 19.

## 4.3 Ablation Study

This section provides two ablations studies, comparing vanilla training versus RTIL for each intermediate layer trained using ILO-RTIL and mGANprior-RTIL. Each experiment was tested on 5 images sampled randomly from CelebA-HQ.

### 4 .3.1   ILO-RTIL

All results refer to Figure 6, recall Figure 3 and Algorithm 1 for notation, $z_0$ refers to optimizing over the initial latent vector (Bora et al., 2017), $z_1$ denotes sequentially optimizing over the first two layers, and so until representation error $z_4$. Refer to the appendix for number of iterations per intermediate layer.

Compressed Sensing - Overall, optimizing up to $z_4$ achieves the best performance for ILO-RTIL and ILO, in most under-sampling ratio's it outperforms and at worst ties compared to other intermediate optimization layers. Comparing ILO-RTIL to ILO, there is an increase in reconstruction performance on average across each under-sampling for $z_4$ of 1.41 dB's, $z_3$ .81 dB's, $z_2$ .4 dB's, and $z_1$ .86 dB's. Inpaiting-Refer to appendix for inpainting results in Figure 6.

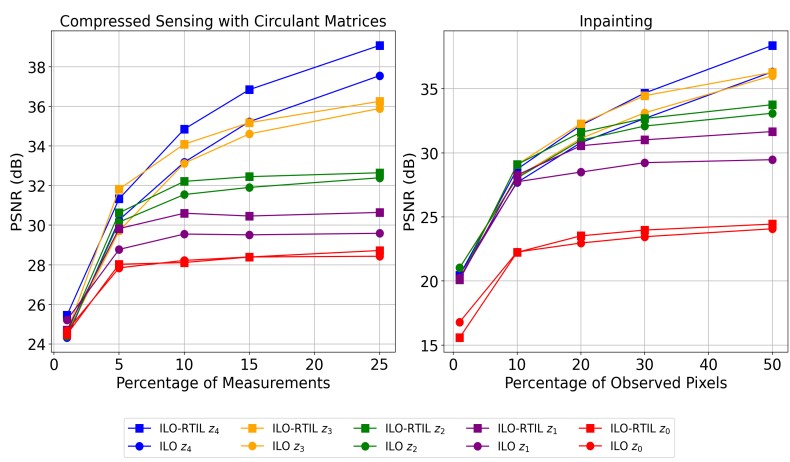

Figure 6: Comparing compressed sensing and inpainting reconstructing performance between ILO-RTIL and ILO for a various number of intermediate layers.

Inpainting - Overall, ILO-RTIL sequentially optimizing up to $z_4$ performs the best overall, outperforming other intermediate representations $z_3, z_2, z_1$ at 50% observed pixels then achieving the slightly better performance at the under-sampling ratio's. Comparing ILO-RTIL to ILO, there is increase in performance on average between each experiment for $z_4$ 1.27 dB's, $z_3$ .55 dB's, $z_2$ .4 dB's, and $z_1$ 1.27 dB's. Moreover, ILO-RTIL from $z_4$ to $z_3$ on average there is increase of performance of .5 dB's, where for vanilla trained there is a decrease of .28 dB's.

### 4 .3.2   mGANprior-RTIL

Compressed Sensing- mGANpior-RTIL outperformed vanilla mGANpior in each optimization setting using $N = \{1, 10, 20\}$, marginal at $N = 1$, but more noticeable at $N = \{10, 20\}$. Overall, $N = 20$ achieves the best performance for both mGANprior and mGANprior-RTIL training across each under-sampling regime. As well as mGANprior-RTIL increases performance on average across all the under-sampling experiments with $N = 10$ compared to $N = 20$ with mGANprior. Inpaiting-Refer to appendix for inpainting results in Figure 7.Overall, $N = 20$ achieves the best performance for both vanilla and RTIL training across each under-sampling regime. For mGANprior has a substantial

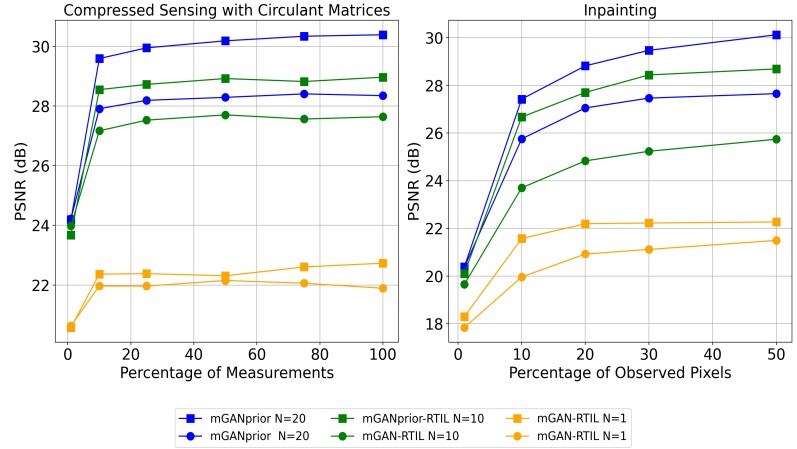

Figure 7: Comparing compressed sensing and inpainting reconstructing performance between mGANprior-RTIL and mGANprior for various number of latent codes

increase in performance (1.83 dB's) between 10 latent codes and 20 on average across each experiment. RTIL training increases performance not as much, but a notable amount with an increase of .93 dBs. However, mGANprior-RTIL with $N = 10$ outperforms mGANprior on average across all the under-sampling ratios of about .66 dBs.

# 5    Theoretical Model for Compressed Sensing with RTIL

In this section, we examine a simple theoretical model for compressive sensing with RTIL. Our objective is to shed some light on the reasons why RTIL may improve the performance of inversion algorithms that optimize over intermediate layers.

We begin with an informal discussion. Recall that for a base generative architecture $G = g_1 \circ g_0$ given by the composition of two computational modules $g_0 : \mathbb{R}^{n_0} \to \mathbb{R}^{n_1}$ and $g_1 : \mathbb{R}^{n_1} \to \mathbb{R}^{n_d}$, ILO uses the extended generative network $\widetilde{G} : \mathbb{R}^{n_0} \times \mathbb{R}^{n_1} \to \mathbb{R}^{n_d}$, such that $\widetilde{G}(\boldsymbol{z}_0, \boldsymbol{z}_1) = g_1(\boldsymbol{z}_1 + g_0(\boldsymbol{z}_0))$.

We would like to compare the performance of an extended model $\widetilde{G}^{\mathrm{Van}}$ trained in a standard/vanilla way and an extended model $\widetilde{G}^{\mathrm{RTIL}}$ trained using RTIL the principle. In the vanilla case, during training, the intermediate layer variable $\boldsymbol{z}_1$ is ignored and the network may be trained via the standard minimax formulation

$$\min_{\theta} \max_{\Theta} \mathbb{E}_{\boldsymbol{x} \sim p_{\boldsymbol{x}}} \big[ \log D_{\Theta}(\boldsymbol{x}) \big] + \mathbb{E}_{\boldsymbol{z}_0 \sim p_{\boldsymbol{z}_0}} \big[ \log(1 - D_{\Theta}(\widetilde{G}_{\theta}^{\mathrm{Van}}(\boldsymbol{z}_0, 0))) \big]. \tag{3}$$

Using the RTIL principle instead the network $\widetilde{G}^{\mathrm{RTIL}}$ may be trained with (taking $\lambda = 0$ in 2 for simplicity)

$$\min_{\theta} \max_{\Theta} \mathbb{E}_{\boldsymbol{x} \sim p_{\boldsymbol{x}}} \big[ \log D_{\Theta}(\boldsymbol{x}) \big] + \mathbb{E}_{\substack{\boldsymbol{z}_0 \sim p_{\boldsymbol{z}_0} \\ \boldsymbol{z}_1 \sim p_{\boldsymbol{z}_1}}} \big[ \log(1 - D_{\Theta}(\widetilde{G}_{\theta}^{\mathrm{RTIL}}(\boldsymbol{z}_0, \boldsymbol{z}_1))) \big]. \tag{4}$$

For compressed measurements $\boldsymbol{y}$ with the sensing matrix $\mathbf{A}$, then ILO would perform the following steps (see Algorithm 1)

$$\begin{aligned} \hat{\boldsymbol{z}}_0 &= \arg \min_{\boldsymbol{z}_0' \in \mathbb{R}^{n_0}} \|\boldsymbol{y} - \mathbf{A}\widetilde{G}(\boldsymbol{z}_0', 0)\|_2^2, \quad \text{initialize } \boldsymbol{z}_0' \sim \mathcal{N}(0, \boldsymbol{I}_{n_0}), \\ \hat{\boldsymbol{z}}_1 &= \arg \min_{\boldsymbol{z}_1' \in \mathbb{R}^{n_1}} \|\boldsymbol{y} - \mathbf{A}\widetilde{G}(\hat{\boldsymbol{z}}_0, \boldsymbol{z}_1')\|_2^2, \quad \text{initialize } \boldsymbol{z}_1' \sim \mathcal{N}(0, \boldsymbol{I}_{n_1}). \end{aligned} \tag{5}$$

where $\widetilde{G} = \widetilde{G}_{\theta}^{\mathrm{Van}}$ or $\widetilde{G} = \widetilde{G}_{\theta}^{\mathrm{RTIL}}$.

Notice now, that when training $\tilde{G}^{\mathrm{Van}}$ using 3 we compute $\tilde{G}^{\mathrm{Van}}(\boldsymbol{z}_0, 0) = g_1(g_0(\boldsymbol{z}_0))$. During training, the computational block $g_1$ therefore only receives input vectors in the range of $g_0$, and is not specified in the directions off the range of $g_0$. During inference, in the second step of 5, though, we compute $\tilde{G}^{\mathrm{Van}}(\hat{\boldsymbol{z}}_0, \boldsymbol{z}_1') = g_1( g_0(\hat{\boldsymbol{z}}_0) + \boldsymbol{z}_1' )$. Hence, ILO optimizes the variable $\boldsymbol{z}_1'$ in directions for which the behavior of $g_1$ may not be well specified, which could lead to large errors.

Contrary to vanilla training, when training $\widetilde{G}_{\theta}^{\mathrm{RTIL}}$ with RTIL we compute $\tilde{G}^{\mathrm{RTIL}}(\boldsymbol{z}_0, \boldsymbol{z}_1) = g_1(g_0(\boldsymbol{z}_0) + \boldsymbol{z}_1)$. Hence, we expect the computational block $g_1$ to learn meaningful outputs for inputs $g_0(\boldsymbol{z}_0) + \boldsymbol{z}_1$ off the range of $g_0$ and ILO to perform better.

In the next sections, we formalize the above argument. To provide the most easily understandable context, we consider the case of a generative model given by a two-layer linear neural network. Further, we consider the case where the generative model is trained in a supervised manner, and we consider the regime of infinite training data. The supervised setting allows us to avoid the technicalities of unsupervised adversarial training, and working in the infinite data regime allows us to avoid statistical estimation errors.

## 5 .1    Problem Setup

We assume that the true signal distribution is given by $\boldsymbol{x} = G^{\star}(\boldsymbol{z}_0, \boldsymbol{z}_1) = \boldsymbol{W}_1^{\star}(\boldsymbol{W}_0^{\star}\boldsymbol{z}_0 + \boldsymbol{z}_1)$, where $G^{\star} : \mathbb{R}^{n_0} \times \mathbb{R}^{n_1} \to \mathbb{R}^{n_d}$, $\boldsymbol{z}_0 \sim \mathcal{N}(0, \boldsymbol{I}_{n_0})$ and $\boldsymbol{z}_1 \sim \mathcal{N}(0, \boldsymbol{I}_{n_1})$ drawn independently, and where $\boldsymbol{W}_1^{\star} \in \mathbb{R}^{n_d \times n_1}$ and $\boldsymbol{W}_0^{\star} \in \mathbb{R}^{n_1 \times n_0}$. We moreover assume that $n_0 < n_1 < n_d$, and $\boldsymbol{W}_0^{\star}, \boldsymbol{W}_1^{\star}$ are full rank. We furthermore assume that $\boldsymbol{W}_0^{\star}$ is known, but that $\boldsymbol{W}_1^{\star}$ is unknown.

Let $\boldsymbol{x}^{\star} = G^{\star}(\boldsymbol{z}_0^{\star}, \boldsymbol{z}_1^{\star})$ be an unknown signal, where $\boldsymbol{z}_0^{\star} \sim \mathcal{N}(0, \boldsymbol{I}_{n_0})$ and $\boldsymbol{z}_1^{\star} \sim \mathcal{N}(0, \boldsymbol{I}_{n_1})$. We consider the problem of recovering $\boldsymbol{x}^{\star}$ given compressed linear measurements $\boldsymbol{y} = \mathbf{A}\boldsymbol{x}^{\star} \in \mathbb{R}^m$ where $n_1 \leq m < n_d$.

We assume that a practitioner has chosen a base generative model $G : \mathbb{R}^k \to \mathbb{R}^{n_d}$ given by $G : \boldsymbol{z}_0 \mapsto G(\boldsymbol{z}_0) = \boldsymbol{W}_1(\boldsymbol{W}_0^{\star}\boldsymbol{z}_0)$. Since this model has a nonzero representation error the practitioner will then consider the extended generative network $\widetilde{G}(\boldsymbol{z}_0, \boldsymbol{z}_1) : \mathbb{R}^{n_0} \times \mathbb{R}^{n_1} \to \mathbb{R}^{n_d}$ given by $\widetilde{G} : \boldsymbol{z}_0, \boldsymbol{z}_1 \mapsto \widetilde{G}(\boldsymbol{z}_0, \boldsymbol{z}_1) = \boldsymbol{W}_1(\boldsymbol{W}_0^{\star}\boldsymbol{z}_0 + \boldsymbol{z}_1)$, and an inversion method that estimates $\boldsymbol{x}^{\star}$ as $\boldsymbol{x}^{\star} \approx \widetilde{G}(\hat{\boldsymbol{z}}_0, \hat{\boldsymbol{z}}_1)$ where $(\hat{\boldsymbol{z}}_0, \hat{\boldsymbol{z}}_1)$ are found using ILO 5

We are then interested in analyzing the effects of different training methods for $\widetilde{G}$ on the estimation of $\boldsymbol{x}^{\star}$ via the inversion method 5.

## 5 .2 Training the Models

We will now compare the training of $\widetilde{G}$ with vanilla training and with RTIL. The model trained with vanilla training will be denoted by $\widetilde{G}^{\text{Van}}$ and its weight by $\boldsymbol{W}_1^{\text{Van}}$. The model trained with RTIL will be denoted by $\widetilde{G}^{\text{RTIL}}$ and its weight by $\boldsymbol{W}_1^{\text{RTIL}}$.

Training each model consists of estimating $\boldsymbol{W}_1^{\text{Van}}$ and $\boldsymbol{W}_1^{\text{RTIL}}$ under a least squares loss. In the idealized regime of infinite training data, vanilla training the generative network $\widetilde{G}^{\text{Van}}$ corresponds to ignoring the intermediate variables $\boldsymbol{z}_1$, and in particular estimating $\boldsymbol{W}_1^{\star}$ by

$$\boldsymbol{W}_1^{\text{Van}} \in \underset{\boldsymbol{W}_1 \in \mathbb{R}^{n_d \times n_1}}{\arg\min} \ \mathbb{E}_{\boldsymbol{z}_0, \boldsymbol{z}_1} \|G^{\star}(\boldsymbol{z}_0, \boldsymbol{z}_1) - \widetilde{G}^{\text{Van}}(\boldsymbol{z}_0, 0)\|_2^2. \tag{6}$$

Since the objective is to solve compressed sensing with the intermediate layer optimization 5, training $\widetilde{G}^{\text{RTIL}}$ with RTIL instead corresponds to

$$\boldsymbol{W}_1^{\text{RTIL}} \in \underset{\boldsymbol{W}_1 \in \mathbb{R}^{n_d \times n_1}}{\arg\min} \ \mathbb{E}_{\boldsymbol{z}_0, \boldsymbol{z}_1} \|G^{\star}(\boldsymbol{z}_0, \boldsymbol{z}_1) - \widetilde{G}^{\text{RTIL}}(\boldsymbol{z}_0, \boldsymbol{z}_1)\|_2^2. \tag{7}$$

The next lemma characterizes the solutions of the optimization problems 6 and 7.

**Lemma 5 .1.** *Let $\boldsymbol{W}_1^{Van}$ as in 6 then $\boldsymbol{W}_1^{Van}\boldsymbol{W}_0^{\star} = \boldsymbol{W}_1^{\star}\boldsymbol{W}_0^{\star}$. Moreover, there exists a unique $\boldsymbol{W}_1^{RTIL}$ solution of 7, given by $\boldsymbol{W}_1^{RTIL} = \boldsymbol{W}_1^{\star}$.*

While the above lemma shows that $\boldsymbol{W}_1^{\text{Van}}$ will equal $\boldsymbol{W}_1^{\star}$ on the range of $\boldsymbol{W}_0^{\star}$, its behavior on the space orthogonal to the range of $\boldsymbol{W}_0^{\star}$ depends on how 6 is solved. To simplify the discussion below, we will consider $\boldsymbol{W}_1^{\text{Van}}$ to be the minimum (Frobenius) norm solution of 6.

## 5 .3 Compressed Sensing

We denote by $(\boldsymbol{z}_0^{\text{Van}}, \boldsymbol{z}_1^{\text{Van}})$ the solutions of the minimization problems 5 using $\widetilde{G}^{\text{Van}}$ in place of $\widetilde{G}$. The estimate of $x^{\star}$ using the vanilla trained model is then $\boldsymbol{x}^{\star} \approx \widetilde{G}^{\text{Van}}(\boldsymbol{z}_0^{\text{Van}}, \boldsymbol{z}_1^{\text{Van}})$. Similarly, $(\boldsymbol{z}_0^{\text{RTIL}}, \boldsymbol{z}_1^{\text{RTIL}})$ are the solutions of 5 using $\widetilde{G}^{\text{RTIL}}$ in place of $\widetilde{G}$, so that the estimate of $x^{\star}$ using the RTIL model is $\boldsymbol{x}^{\star} \approx \widetilde{G}^{\text{Van}}(\boldsymbol{z}_0^{\text{RTIL}}, \boldsymbol{z}_1^{\text{RTIL}})$.

The following lemma quantifies the reconstruction errors made by using the two generative models.

**Lemma 5 .2.** *Assume that $\boldsymbol{W}_0^{\star}, \boldsymbol{W}_1^{\star}$ are full rank. Let $\boldsymbol{W}_1^{Van}$ be the minimum norm solution of 6 and $\boldsymbol{W}_1^{RTIL}$ be the solution of 7. Let $\mathbf{A} \in \mathbb{R}^{m \times n_d}$ with i.i.d. $\mathcal{N}(0,1)$ entries. Then with probability 1*

$$\mathbb{E}_{\boldsymbol{z}_0^{\star}, \boldsymbol{z}_1^{\star}}\left[\|G^{\star}(\boldsymbol{z}_0^{\star}, \boldsymbol{z}_1^{\star}) - \widetilde{G}^{Van}(\boldsymbol{z}_0^{Van}, \boldsymbol{z}_1^{Van})\|^2\right] \geq \max_{h \in range(\boldsymbol{W}_0^{\star})^{\perp}} \|(\boldsymbol{I}_{n_d} - \mathcal{P}_{\boldsymbol{W}_1^{\star}\boldsymbol{W}_0^{\star}})\boldsymbol{W}_1^{\star}h\|_2^2 > 0 \tag{8}$$

*and*

$$\mathbb{E}_{\boldsymbol{z}_0^{\star}, \boldsymbol{z}_1^{\star}}[\|G^{\star}(\boldsymbol{z}_0^{\star}, \boldsymbol{z}_1^{\star}) - \widetilde{G}^{RTIL}(\boldsymbol{z}_0^{RTIL}, \boldsymbol{z}_1^{RTIL})\|^2] = 0. \tag{9}$$

The above results illustrate why training generative models using RTIL can enable better compressed sensing performance. In the simplified setting of a two-layer linear neural network trained in a supervised manner

with a known first layer and in the infinite data regime, we see that the generative model trained with vanilla training incurs error in the second layer's weights in the orthogonal complement of the range of the first layer. This results in an increase in the reconstruction error when solving compressed sensing.

## 6 Conclusion

We have introduced a principle for training GANs that are intended to be used for solving inverse problems. That principle states that if the inversion algorithm optimizes over intermediate layers of the network, then during training the network should be regularized in those layers. We instantiate this principle for two recent and successful optimization algorithms, Intermediate Layer Optimization (Daras et al., 2021) and the Multi-Code Prior (Gu et al., 2020). For both of these algorithms, we devise a new GAN training algorithm. Empirically, we show our trained GANs allow better reconstruction in compressed sensing, inpainting, and super-resolution across multiple under-sampling regimes when compared to GANs trained in a vanilla manner. We note that our methodology only applies in the case of inversion methods that optimize over intermediate layers. However, there have been multiple competitive methods of this form published recently, each of which we show can benefit from this approach. Tools like those proposed in this paper, provided sufficient computational resources, may allow these methods to be even more competitive in real-world applications. verification at larger image sizes and on generative models trained specifically for certain applications, such as MRI can benefit from the increase in performance.

## 7 Acknowledgements and Disclosure of Funding

PH acknowledges support from NSF Awards DMS-2053448, DMS-1848087, and DMS-2022205.

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

# A    Proof

*Proof of Lemma 5.1.* Notice that for any $\boldsymbol{W}_1 \in \mathbb{R}^{n_d \times n_1}$

$$
\mathbb{E}_{\boldsymbol{z}_0, \boldsymbol{z}_1}[\|\boldsymbol{W}_1^\star(\boldsymbol{W}_0^\star \boldsymbol{z}_0 + \boldsymbol{z}_1) - \boldsymbol{W}_1 \boldsymbol{W}_0^\star \boldsymbol{z}_0\|_2^2] = \mathbb{E}_{\boldsymbol{z}_0}[\|(\boldsymbol{W}_1^\star - \boldsymbol{W}_1)\boldsymbol{W}_0^\star \boldsymbol{z}_0\|_2^2] + \mathbb{E}_{\boldsymbol{z}_1}[\|\boldsymbol{W}_1^\star \boldsymbol{z}_1\|_2^2]
$$
$$
= \|(\boldsymbol{W}_1^\star - \boldsymbol{W}_1)\boldsymbol{W}_0^\star\|_F^2 + \|\boldsymbol{W}_1^\star\|_F^2,
$$

where the first equality used independence of $\boldsymbol{z}_0$ and $\boldsymbol{z}_1$ and the second used the fact that $\mathbb{E}_{\boldsymbol{z} \sim \mathcal{N}(0, \boldsymbol{I}_n)}\|\boldsymbol{M}\boldsymbol{z}\|_2^2 = \|\boldsymbol{M}\|_F^2$ for any matrix $\boldsymbol{M} \in \mathbb{R}^{m \times n}$. The solutions of equation 6 are then solutions of

$$
\min_{\boldsymbol{W}_1 \in \mathbb{R}^{n_1 \times n_0}} \|\mathcal{L}(\boldsymbol{W}_1^\star) - \mathcal{L}(\boldsymbol{W}_1)\|_F^2 \tag{10}
$$

where $\mathcal{L} : \mathbb{R}^{n_d \times n_1} \to \mathbb{R}^{n_1 \times n_0}$ is the linear operator given by $\mathcal{L} : \boldsymbol{W}_1 \mapsto \boldsymbol{W}_1 \boldsymbol{W}_0^\star$. Since $n_1 > n_0$, this operator is singular and there are infinite solutions of equation 10, all of which satisfy $\mathcal{L}(\boldsymbol{W}_1^\star) = \mathcal{L}(\boldsymbol{W}_1)$, i.e. the thesis.

Regarding 7, observe instead that for any $W_1 \in \mathbb{R}^{n_d \times n_1}$

$$
\mathbb{E}_{\boldsymbol{z}_0, \boldsymbol{z}_1}[\|\boldsymbol{W}_1^\star(\boldsymbol{W}_0^\star \boldsymbol{z}_0 + \boldsymbol{z}_1) - \boldsymbol{W}_1(\boldsymbol{W}_0^\star \boldsymbol{z}_0 + \boldsymbol{z}_1)\|_2^2] = \|(\boldsymbol{W}_1^\star - \boldsymbol{W}_1)\boldsymbol{W}_0^\star\|_F^2 + \|\boldsymbol{W}_1^\star - \boldsymbol{W}_1\|_F^2.
$$

This shows $\boldsymbol{W}_1 = \boldsymbol{W}_1^\star$ is the unique minimizer of 7. $\qquad\square$

Before analyzing the solution of compressed sensing with the trained generative models, we observe the following fact on the minimum norm solution of 6.

**Lemma A.1.** *Let $\boldsymbol{W}_1^{Van}$ be the minimum Frobenius norm solution of equation 6. Then $\boldsymbol{W}_1^{Van} = \boldsymbol{W}_1^\star \boldsymbol{W}_0^\star (\boldsymbol{W}_0^\star)^\dagger$ where $(\boldsymbol{W}_0^\star)^\dagger$ is the pseudoinverse of $\boldsymbol{W}_0^\star$.*

*Proof.* Notice that $\mathcal{L}(\boldsymbol{W}_1^\star) = \mathcal{L}(\boldsymbol{W}_1^{\mathrm{Van}})$, where $\mathcal{L}$ is defined in the proof of Lemma equation 5.1. We next show that $\boldsymbol{W}_1^{\mathrm{Van}}$ is orthogonal to the null space of $\mathcal{L}$, which implies the thesis.

Let $\boldsymbol{W}_1 \in \mathbb{R}^{n_d \times n_1}$ be such that $\mathcal{L}(\boldsymbol{W}_1) = \boldsymbol{W}_1 \boldsymbol{W}_0^\star = 0$. Then we have

$$
\begin{aligned}
\langle \boldsymbol{W}_1^{\mathrm{Van}}, \boldsymbol{W}_1 \rangle_F &= tr((\boldsymbol{W}_1^{\mathrm{Van}})^T \boldsymbol{W}_1) \\
&= tr(\boldsymbol{W}_1(\boldsymbol{W}_0^\star(\boldsymbol{W}_0^\star)^\dagger)^T (\boldsymbol{W}_1^\star)^T) \\
&= tr(\boldsymbol{W}_1 \boldsymbol{W}_0^\star(\boldsymbol{W}_0^\star)^\dagger (\boldsymbol{W}_1^\star)^T) \\
&= 0,
\end{aligned}
$$

where third equality uses the fact that $\boldsymbol{W}_0^\star(\boldsymbol{W}_0^\star)^\dagger$ is symmetric and the fourth equality uses the assumption on $\boldsymbol{W}_1$. $\qquad\square$

*Proof of Lemma 5 .2.*

- Proof of equation 8. Notice that by the previous lemma $\boldsymbol{W}_1^{\mathrm{Van}} = \boldsymbol{W}_1^\star \boldsymbol{W}_0^\star (\boldsymbol{W}_0^\star)^\dagger = \boldsymbol{W}_1^\star \mathcal{P}_{\boldsymbol{W}_0^\star}$ where $\mathcal{P}_{\boldsymbol{W}_0^\star}$ is the orthogonal projector onto the range of $\boldsymbol{W}_0^\star$. Moreover, with probability 1, $(\mathbf{A}\boldsymbol{W}_1^\star \boldsymbol{W}_0^\star)$ is full rank. It follows that $(\boldsymbol{z}_0^{\mathrm{Van}}, \boldsymbol{z}_1^{\mathrm{Van}})$ is given by

$$\boldsymbol{z}_0^{\mathrm{Van}} = \arg\min_{\boldsymbol{z}_0' \in \mathbb{R}^{n_0}} \|\boldsymbol{y} - \mathbf{A}\boldsymbol{W}_1^\star \boldsymbol{W}_0^\star \boldsymbol{z}_0'\|_2^2,$$

$$\boldsymbol{z}_1^{\mathrm{Van}} = \arg\min_{\boldsymbol{z}_1' \in \mathbb{R}^{n_1}} \|\boldsymbol{y} - \mathbf{A}\boldsymbol{W}_1^\star (\boldsymbol{W}_0^\star \boldsymbol{z}_0^{\mathrm{Van}} + \mathcal{P}_{\boldsymbol{W}_0^\star}\boldsymbol{z}_1')\|_2^2.$$

In particular $\mathcal{P}_{\boldsymbol{W}_0^\star}\boldsymbol{z}_1^{\mathrm{Van}} = 0$ and $(\boldsymbol{I}_{n_1} - \mathcal{P}_{\boldsymbol{W}_0^\star})\boldsymbol{z}_1^{\mathrm{Van}}$ is fixed at initialization. We have then $\widetilde{G}^{\mathrm{Van}}(\boldsymbol{z}_0^{\mathrm{Van}}, \boldsymbol{z}_1^{\mathrm{Van}}) = \boldsymbol{W}_1^\star(\boldsymbol{W}_0^\star \boldsymbol{z}_0^\star + \boldsymbol{W}_0^\star \mathcal{M}_1 \boldsymbol{z}_1^\star)$ where $\mathcal{M}_1 = (A\boldsymbol{W}_1^\star \boldsymbol{W}_0^\star)^\dagger A \boldsymbol{W}_1^\star$.

The reconstruction error is then given by

$$\mathbb{E}_{\boldsymbol{z}_0^\star, \boldsymbol{z}_1^\star} \|G^\star(\boldsymbol{z}_0^\star, \boldsymbol{z}_1^\star) - \widetilde{G}^{\mathrm{Van}}(\boldsymbol{z}_0^{\mathrm{Van}}, \boldsymbol{z}_1^{\mathrm{Van}})\|_2^2 = \|\boldsymbol{W}_1^\star - \boldsymbol{W}_1^\star \boldsymbol{W}_0^\star \mathcal{M}_1\|_F^2$$

$$\geq \min_{\mathcal{M} \in \mathbb{R}^{n_0 \times n_1}} \|\boldsymbol{W}_1^\star - \boldsymbol{W}_1^\star \boldsymbol{W}_0^\star \mathcal{M}\|_F^2$$

$$= \|(\boldsymbol{I}_{n_d} - \mathcal{P}_{\boldsymbol{W}_1^\star \boldsymbol{W}_0^\star})\boldsymbol{W}_1^\star\|_F^2$$

where the first equality follows from the properties of the normal distribution. Regarding then the minimization problem $\min_{\mathcal{M} \in \mathbb{R}^{n_0 \times n_1}} \|\boldsymbol{W}_1^\star - \boldsymbol{W}_1^\star \boldsymbol{W}_0^\star \mathcal{M}_1\|_F^2$, notice that this is convex and the critical points satisfy

$$(\boldsymbol{W}_1^\star \boldsymbol{W}_0^\star)^T \boldsymbol{W}_1^\star = (\boldsymbol{W}_1^\star \boldsymbol{W}_0^\star)^T (\boldsymbol{W}_1^\star \boldsymbol{W}_0^\star)\mathcal{M}$$

Using the fact that $\boldsymbol{W}_1^\star \boldsymbol{W}_0^\star$ is full rank, the unique solution is found to be $[(\boldsymbol{W}_1^\star \boldsymbol{W}_0^\star)^T (\boldsymbol{W}_1^\star \boldsymbol{W}_0^\star)]^{-1}(\boldsymbol{W}_1^\star \boldsymbol{W}_0^\star)^T \boldsymbol{W}_1^\star$, which gives the last equality.

Note now that $\mathcal{P}_{\boldsymbol{W}_1^\star \boldsymbol{W}_0^\star}$ is the projector onto the range of $\boldsymbol{W}_1^\star \boldsymbol{W}_0^\star$ and $\boldsymbol{W}_1^\star$ is full rank. Thus

$$\|(\boldsymbol{I}_{n_d} - \mathcal{P}_{\boldsymbol{W}_1^\star \boldsymbol{W}_0^\star})\boldsymbol{W}_1^\star\|_F^2 \geq \|(\boldsymbol{I}_{n_d} - \mathcal{P}_{\boldsymbol{W}_1^\star \boldsymbol{W}_0^\star})\boldsymbol{W}_1^\star\|_2^2 = \max_{h \in \mathrm{range}(\boldsymbol{W}_0^\star)^\perp} \|(\boldsymbol{I}_{n_d} - \mathcal{P}_{\boldsymbol{W}_1^\star \boldsymbol{W}_0^\star})\boldsymbol{W}_1^\star h\|_2^2 > 0.$$

- *Proof of equation 9* Notice again that with probability 1, $A\boldsymbol{W}_1^\star \boldsymbol{W}_0^\star$ and $A\boldsymbol{W}_1^\star$ have full rank. Moreover $\widetilde{G}^{\mathrm{RTIL}}(\boldsymbol{z}_0^{\mathrm{RTIL}}, \boldsymbol{z}_1^{\mathrm{RTIL}}) = \boldsymbol{W}_1^\star(\boldsymbol{W}_0^\star \boldsymbol{z}_0^{\mathrm{RTIL}} + \boldsymbol{z}_1^{\mathrm{RTIL}})$ where

$$\boldsymbol{z}_0^{\mathrm{RTIL}} = \arg\min_{\boldsymbol{z}_0' \in \mathbb{R}^{n_0}} \|\boldsymbol{y} - \mathbf{A}\boldsymbol{W}_1^{\mathrm{RTIL}} \boldsymbol{W}_0^\star \boldsymbol{z}_0'\|_2^2,$$

$$\boldsymbol{z}_1^{\mathrm{RTIL}} = \arg\min_{\boldsymbol{z}_1' \in \mathbb{R}^{n_1}} \|\boldsymbol{y} - \mathbf{A}\boldsymbol{W}_1^{\mathrm{RTIL}}(\boldsymbol{W}_0^\star \boldsymbol{z}_0^{\mathrm{RTIL}} + \boldsymbol{z}_1')\|_2^2.$$

It is then easy to see that

$$\boldsymbol{z}_0^{\mathrm{RTIL}} = \boldsymbol{z}_0^\star + (A\boldsymbol{W}_1^\star \boldsymbol{W}_0^\star)^\dagger A \boldsymbol{W}_1^\star \boldsymbol{z}_1^\star$$

$$\boldsymbol{z}_1^{\mathrm{RTIL}} = \boldsymbol{z}_1^\star - \boldsymbol{W}_0^\star (A\boldsymbol{W}_1^\star \boldsymbol{W}_0^\star)^\dagger A \boldsymbol{W}_1^\star \boldsymbol{z}_1^\star$$

where $(\mathbf{A}\boldsymbol{W}_1^\star \boldsymbol{W}_0^\star)^\dagger$ denotes the pseudoinverse of $(\mathbf{A}\boldsymbol{W}_1^\star \boldsymbol{W}_0^\star)$. It then follows that $\boldsymbol{W}_1^\star(\boldsymbol{W}_0^\star \boldsymbol{z}_0^{\mathrm{RTIL}} + \boldsymbol{z}_1^{\mathrm{RTIL}}) = \boldsymbol{W}_1^\star(\boldsymbol{W}_0^\star \boldsymbol{z}_0^\star + \boldsymbol{z}_1^\star) = \boldsymbol{x}^\star$, which implies the thesis. $\square$

# A Appendix

Code provided in supplementary folder. Computational requirements for this paper are two NVIDIA 2080 Ti GPU: training StyleGAN2 uses both GPU's and one GPU for inversion, training PGGAN requires one GPU and one for inversion.

## A.1 RTIL-ILO Training Details

All experiments for ILO inversion method used StyleGAN2 architecture (Karras et al., 2020), both vanilla and RTIL models were trained for 500,000 iterations using the same training parameters, i.e., learning rate, batch size, and regularization updates. Below table 1 outlines the macroscopic view of the StlyeGAN2 architecture. Please refer StlyeGAN2 paper (Karras et al., 2020) for more architecture details, between each convolutional layer there is a normalization operation called weight demodulation. During training for RTIL the distribution was induced after each block in the network, which corresponds to cells 2-6 in table 1 up to 4-th convolutional block.

All experiments for ILO inversion method used StyleGAN2 architecture (Karras et al., 2020), both vanilla and RTIL models were trained for 500,000 iterations using the same training parameters, i.e., learning rate, batch size, and regularization updates. Please refer to the code for more details on the training process and the StlyeGAN2 paper (Karras et al., 2020) for more details on the architecture.

Below Table 1 outlines the macroscopic view of the StlyeGAN2 architecture, between each convolutional layer there is a normalization operation called weight demodulation. During training for RTIL the additional latent variables were added after each block in the network, which correspond to cells 2-6 in Table 1 up to 4-th convolutional block.

Table 1: StyleGan2 for image size $256 \times 256 \times 3$

| Generator | | |
|---|---|---|
| Operation | Activation | Output Shape |
| Latent Vector | None | $512 \times 1 \times 1$ |
| $8\times$ MLP (Mapping Network) | LRelu | $512 \times 14$ |
| Constant input | None | $512 \times 4 \times 4$ |
| Conv $3 \times 3$ | LRelu | $256 \times 4 \times 4$ |
| Upsample | None | $256 \times 8 \times 8$ |
| Conv $3 \times 3$ | LRelu | $256 \times 8 \times 8$ |
| Conv $3 \times 3$ | LRelu | $256 \times 8 \times 8$ |
| Upsample | None | $256 \times 16 \times 16$ |
| Conv $3 \times 3$ | LRelu | $256 \times 16 \times 16$ |
| Conv $3 \times 3$ | LRelu | $256 \times 16 \times 16$ |
| Upsample | None | $256 \times 32 \times 32$ |
| Conv $3 \times 3$ | LRelu | $256 \times 32 \times 32$ |
| Conv $3 \times 3$ | LRelu | $256 \times 32 \times 32$ |
| Upsample | None | $256 \times 64 \times 64$ |
| Conv $3 \times 3$ | LRelu | $256 \times 64 \times 64$ |
| Conv $3 \times 3$ | LRelu | $256 \times 64 \times 64$ |
| Upsample | None | $128 \times 128 \times 128$ |
| Conv $3 \times 3$ | LRelu | $128 \times 128 \times 128$ |
| Conv $3 \times 3$ | LRelu | $128 \times 128 \times 128$ |
| Upsample | None | $128 \times 256 \times 256$ |
| Conv $3 \times 3$ | LRelu | $64 \times 256 \times 256$ |
| Conv $3 \times 3$ | LRelu | $64 \times 256 \times 256$ |
| Conv $1 \times 1$ | Linear | $3 \times 256 \times 256$ |
| **Trainable Parameters** : 12,300,877 | | |

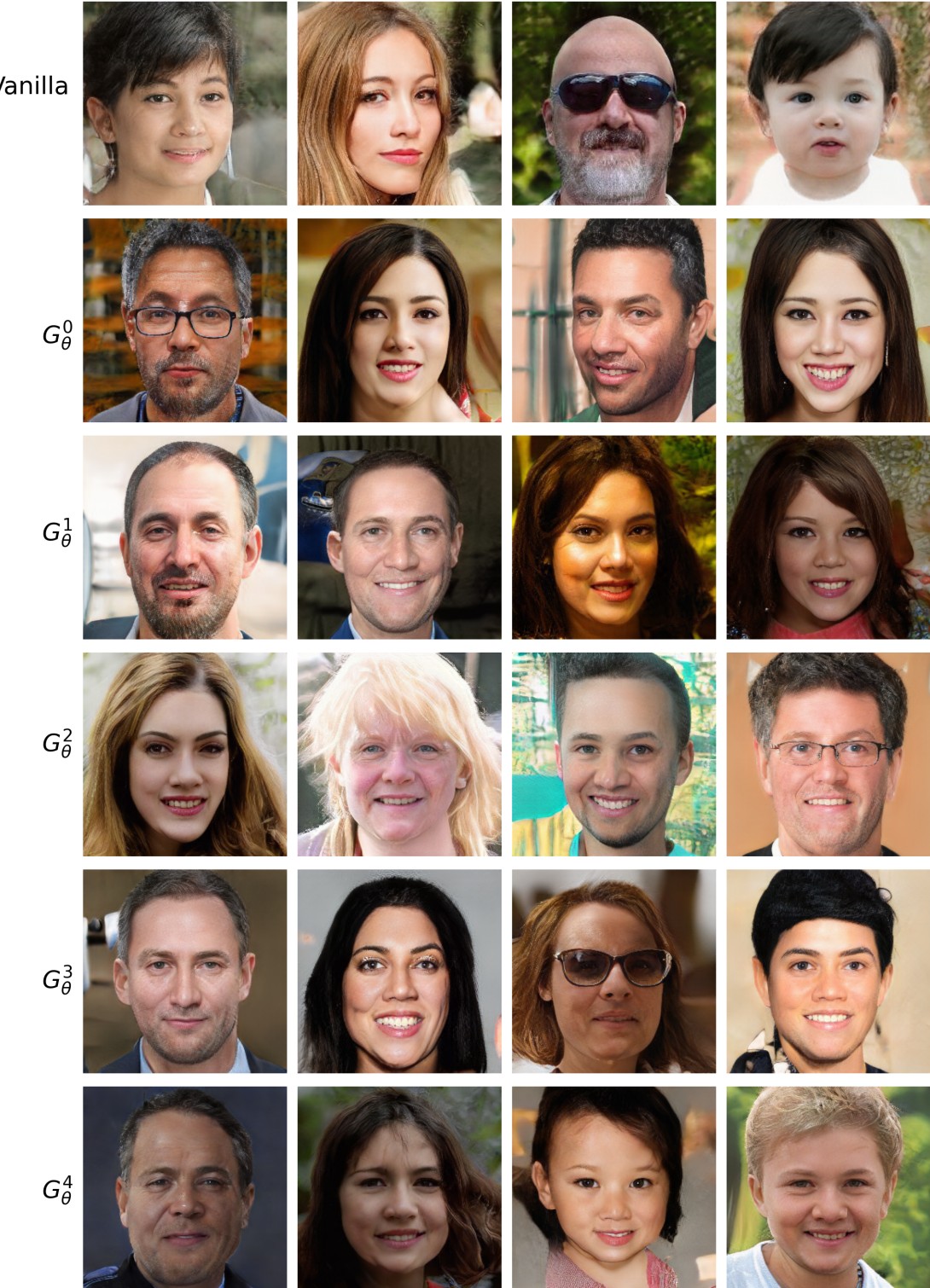

Figure 8: Samples from each of our trained generative models. Top row consist samples from vanilla trained StyleGAN2, then rows $G_\theta^0 \cdots G_\theta^4$ are samples from family of generative models trained with RTIL.

## A.2 RTIL-mGANprior Training Details

All experiments for mGANprior inversion method used PGGAN Architecture (Karras et al., 2018), both vanilla and RTIL models were trained for 600,000 iterations and use the same training parameters, i.e., learning rate, batch size, and regularization updates. If interested, please to refer to the code for more details. Below, table 2 outlines the details of the PGGAN architecture. The intermediate latent variable were added after the 4-th block, which corresponds to the 4-th cell in Table 2.

Table 2: PGGAN for image size $256 \times 256 \times 3$

| Generator | | |
|---|---|---|
| Operation | Activation | Output Shape |
| Latent Vector | None | $512 \times 1 \times 1$ |
| Conv $4 \times 4$ | LRelu | $256 \times 4 \times 4$ |
| Conv $3 \times 3$ | LRelu | $256 \times 4 \times 4$ |
| Upsample | None | $256 \times 8 \times 8$ |
| Conv $3 \times 3$ | LRelu | $256 \times 8 \times 8$ |
| Conv $3 \times 3$ | LRelu | $256 \times 8 \times 8$ |
| Upsample | None | $256 \times 16 \times 16$ |
| Conv $3 \times 3$ | LRelu | $256 \times 16 \times 16$ |
| Conv $3 \times 3$ | LRelu | $256 \times 16 \times 16$ |
| Upsample | None | $256 \times 32 \times 32$ |
| Conv $3 \times 3$ | LRelu | $256 \times 32 \times 32$ |
| Conv $3 \times 3$ | LRelu | $256 \times 32 \times 32$ |
| Upsample | None | $256 \times 64 \times 64$ |
| Conv $3 \times 3$ | LRelu | $128 \times 64 \times 64$ |
| Conv $3 \times 3$ | LRelu | $128 \times 64 \times 64$ |
| Upsample | None | $128 \times 128 \times 128$ |
| Conv $3 \times 3$ | LRelu | $64 \times 128 \times 128$ |
| Conv $3 \times 3$ | LRelu | $64 \times 128 \times 128$ |
| Upsample | None | $64 \times 256 \times 256$ |
| Conv $3 \times 3$ | LRelu | $64 \times 256 \times 256$ |
| Conv $3 \times 3$ | LRelu | $64 \times 256 \times 256$ |
| Conv $1 \times 1$ | Linear | $3 \times 256 \times 256$ |
| **Trainable Parameters** : 7,445,443 | | |

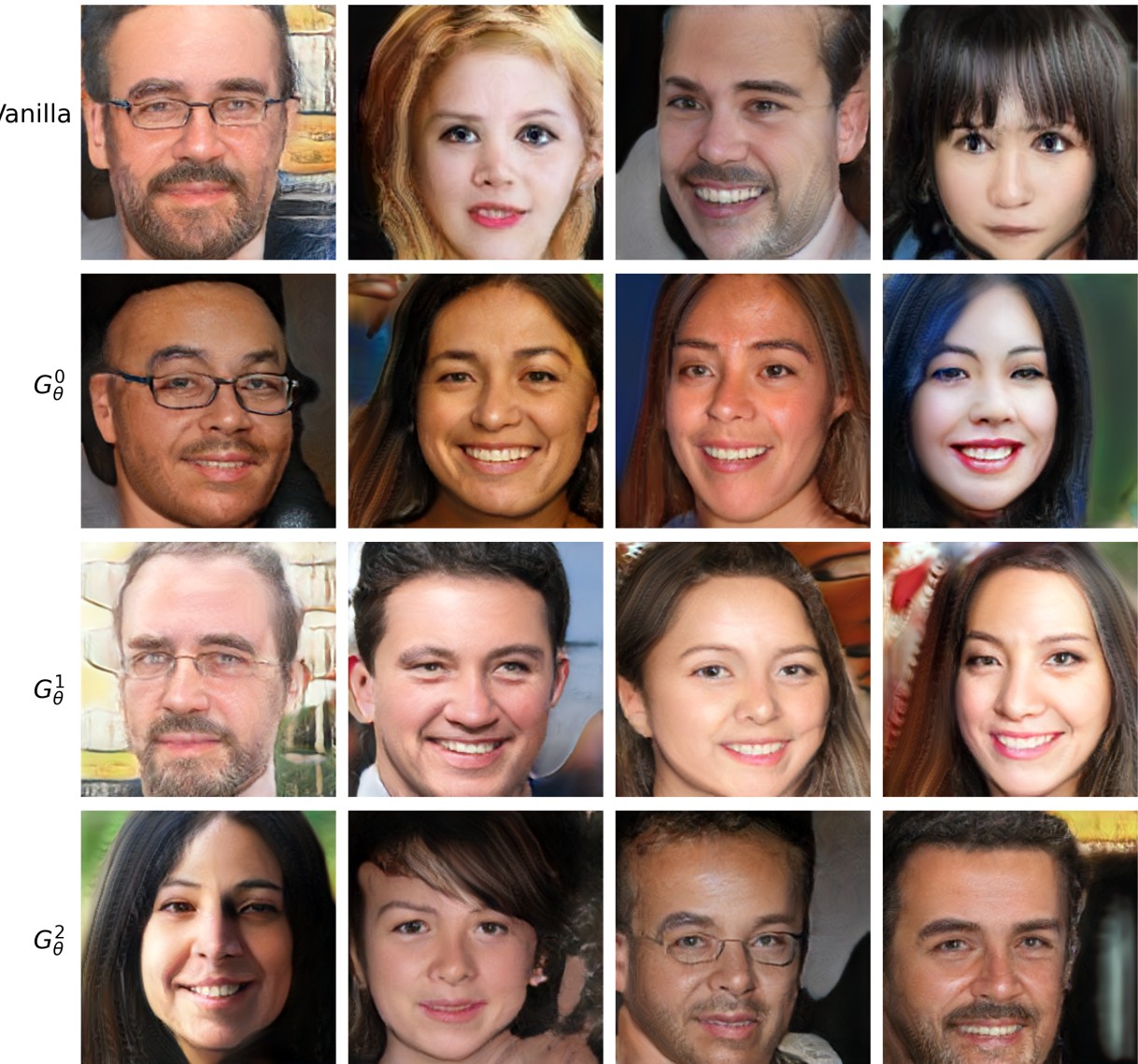

Figure 10: Samples from each of our trained generative models. Top row consist samples from vanilla trained PGGan, then rows $G_\theta^0 \cdots G_\theta^2$ are samples from family of generative models trained with RTIL.

### A.3 Inversion Details for ILO

Hyper-parameters were tuned based on experiments in the appendix of the ILO paper (Daras et al., 2021) and the official GitHub repository. Our implementation uses the code from the GitHub repository, please refer to supplementary material to see. For more details about our hyper-parameter choice, learning rate and number of iterations per layer were tuned for compressed sensing, then the same configuration were used for inpainting and super-resolution. For experiments in Section 4.1 Figure 4 the configuration for the number of iterations per layer is $\{2000, 1000, 1000, 1000, 2000\}$. ILO-RTIL uses a learning rate that begins at .2 at the initial layer, then ramps up linearly and ramps down using a cosine scheduler, as proposed by (Karras et al., 2020). As for ILO, each layer initialized with a learning rate of .1 then optimized independently using the same learning rate scheduler, which is proposed in official Github repository (Daras et al., 2021). We choose the intermediate layer up to which optimize to, based on ablation study Section 4.3.1 Figure 6. Below we

report the loss function used for each inverse problem in case of ILO-RTIL and RTIL proposed in Daras et al. (2021):

- *Compressed Sensing* - Mean square error

- *Inpainting*- Equal weighted combination of mean square error and LPIPS (Zhang et al., 2018) for sufficient number of measurements. Notice that with more than 50% missing pixels LPIPS did not help reconstruction performance.

- *Super-Resolution* - Equal weighted combination of mean square error and LPIPS for ILO-RTIL, for vanilla ILO LPIPS loss is weighted more with $\lambda = 1.5$ . Except at downsampling ratio of $\frac{1}{16}$, only MSE loss is utilized.

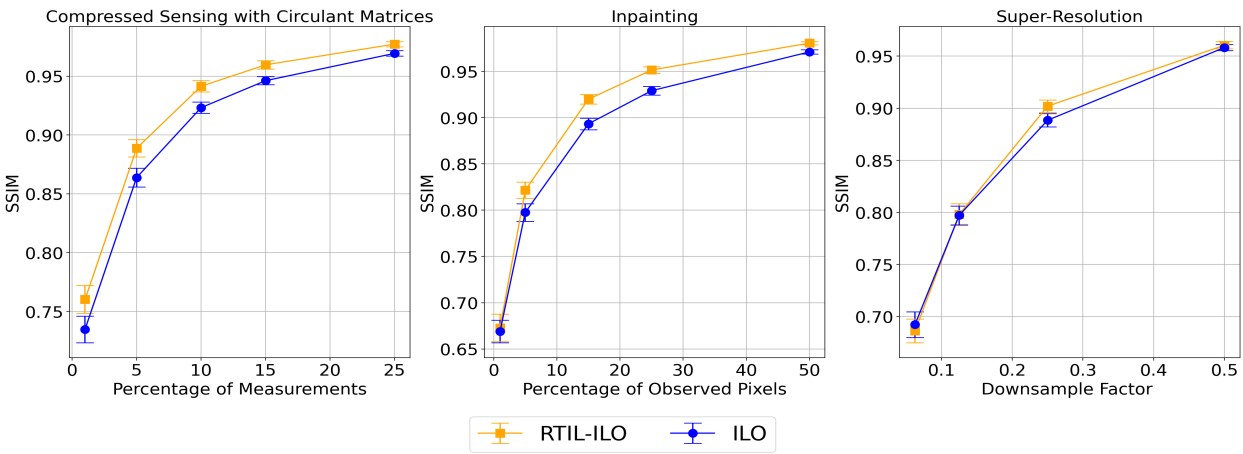

Figure 11: Performance of ILO-RTIL and vanilla trained ILO for Compressed sensing, inpainting, and super resolution for various under-sampling ratios. ILO-RTIL increases performances or ties for each under-sampling ratio with respect to SSIM across each of the inverse problems compared to ILO. The vertical bars indicate 95% confidence intervals.

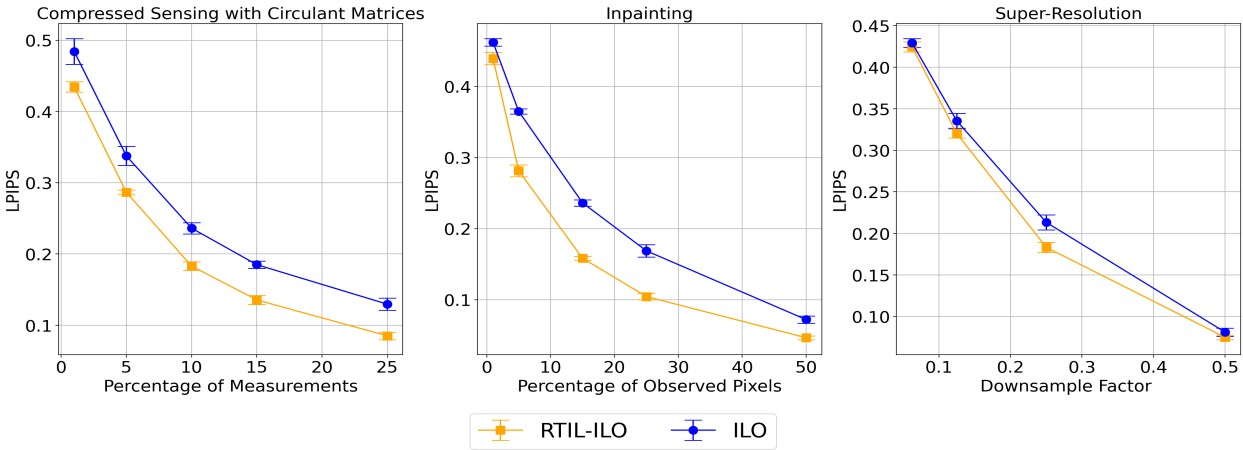

Figure 12: Performance of ILO-RTIL and vanilla trained ILO for Compressed sensing, inpainting, and super resolution for various under-sampling ratios. ILO-RTIL increases performances or ties for each under-sampling ratio with respect to LPIPS across each of the inverse problems compared to ILO. The vertical bars indicate 95% confidence intervals.

### A.4 Inversion Details for mGANprior

Hyper-parameters were tuned based on the Github repository (Gu et al., 2020) for Section 4 .2 Figure 15, learning rate and the number of iterations were tuned for compressed sensing then reused for inpainting and super-resolution. mGANprior-RTIL uses Adam (Kingma & Ba, 2015) optimizer initialized at learning rate of .1 and optimized for 2500 iterations. mGANprior uses SGD initialized at learning rate 1 and optimized for 2500 iterations, which is based on the official GitHub repository (Gu et al., 2020). Empirically, the mGANprior (Gu et al., 2020) improvement in reconstruction saturated after $N = 20$ latent codes. Moreover, for selecting which intermediate layer to optimize over for the vanilla model was determined by the ablation study in Figure 22. Below we report the loss function used for each inverse problem in case of mGANprior-RTIL and mGANprior

- *Compressed Sensing* - Mean square error loss for both methods.

- *Inpainting*- Mean square error plus $l_1$ LPIPS regularization proposed by (Gu et al., 2020). For mGANprior-RTIL the regularization term was $\lambda = .1$ and for mGANprior $\lambda = .5$.

- *Super-Resolution* - Mean square error plus $l_1$ LPIPS regularization proposed by (Gu et al., 2020) or mGANprior-RTIL the regularization term was scaled $\lambda = .1$ and for mGANprior $\lambda = .5$. Except at downsampling ratio of $\frac{1}{16}$ where the we only use MSE loss.

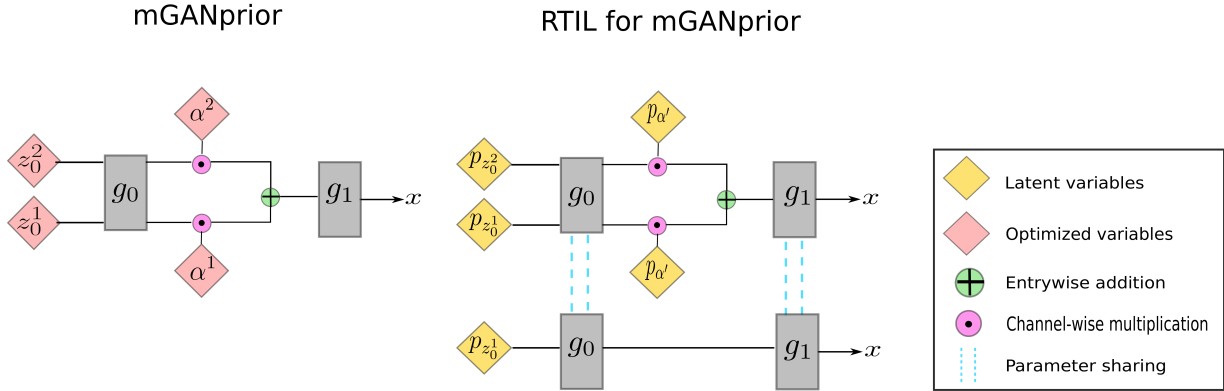

Figure 13: The left side portrays vanilla training for mGANprior, and the right side demonstrates RTIL. For mGANprior-RTIL this example with $N = 2$ latent codes, where the top model trains a vanilla model analogous to model on the left, then parameter shares with the model below.

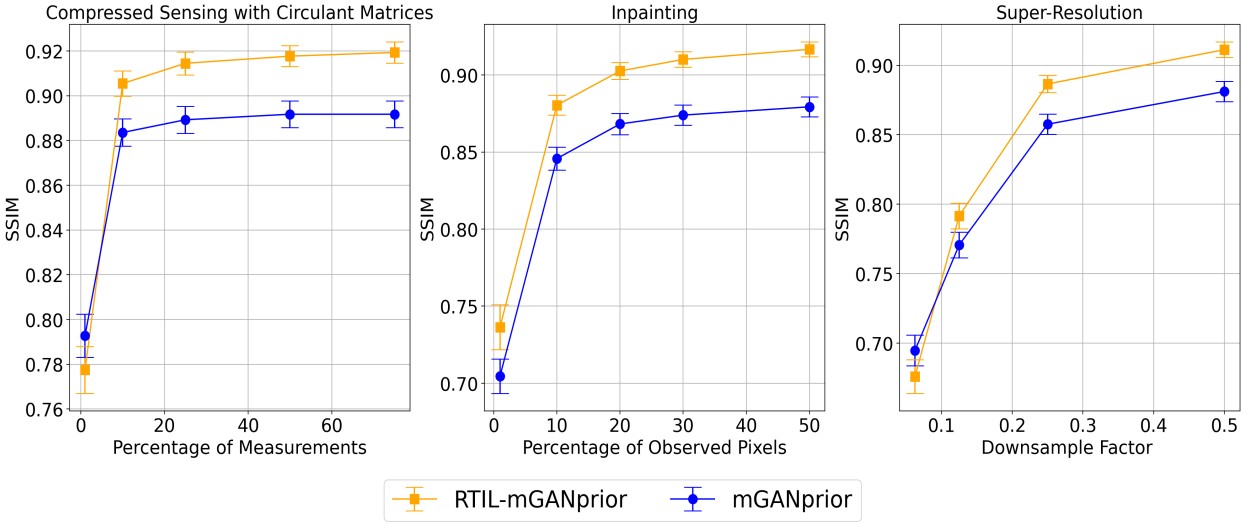

Figure 14: Performance of mGANprior-RTIL and vanilla trained mGANprior for compressed sensing, inpainting, and super resolution for various under-sampling ratios. mGANprior-RTIL increases performances over vanilla mGANprior with respect to SSIM over each under-sampling ratio, except for super-resolution problems at low under-sampling ratio's. The vertical bars indicate 95% intervals.

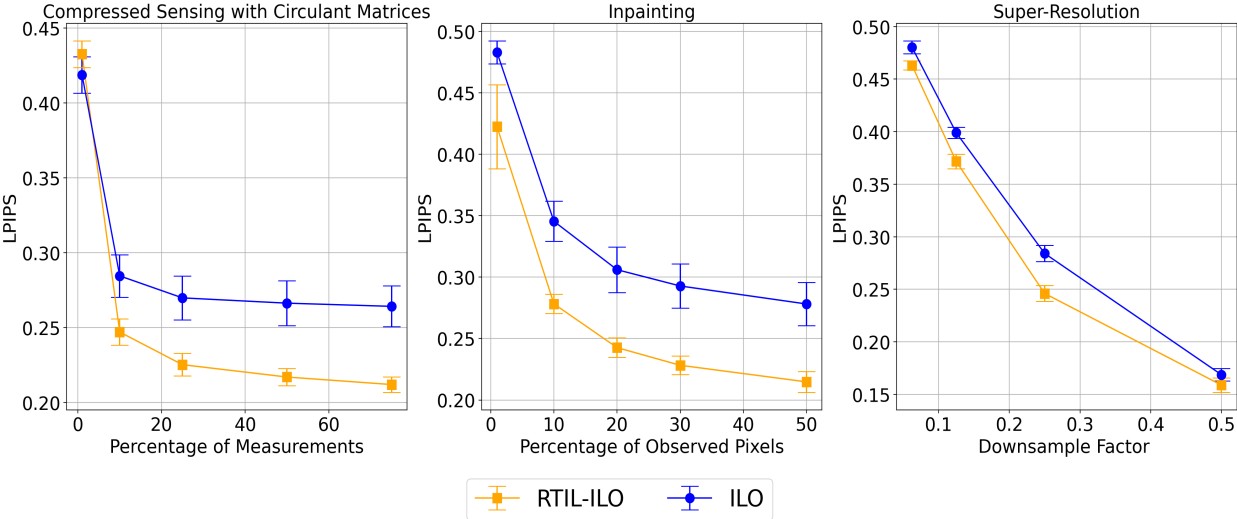

Figure 15: Performance of mGANprior-RTIL and vanilla trained mGANprior for compressed sensing, inpainting, and super resolution for various under-sampling ratios. mGANprior-RTIL increases performances over vanilla mGANprior with respect to SSIM over each under-sampling ratio, except for super-resolution problems at low under-sampling ratio's. The vertical bars indicate 95% intervals.

## A.5 Inverse Problems Qualitative Results

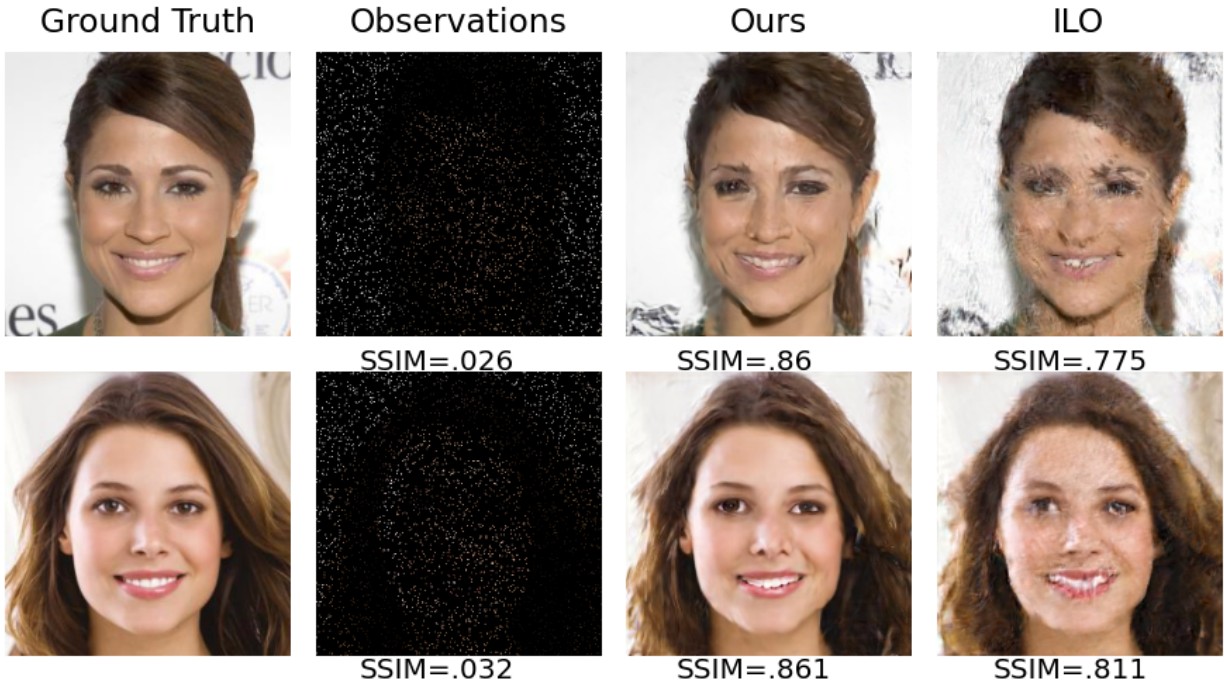

Figure 16: Qualitative comparison between our method ILO-RTIL and ILO for inpainting at 5% of observed pixels.

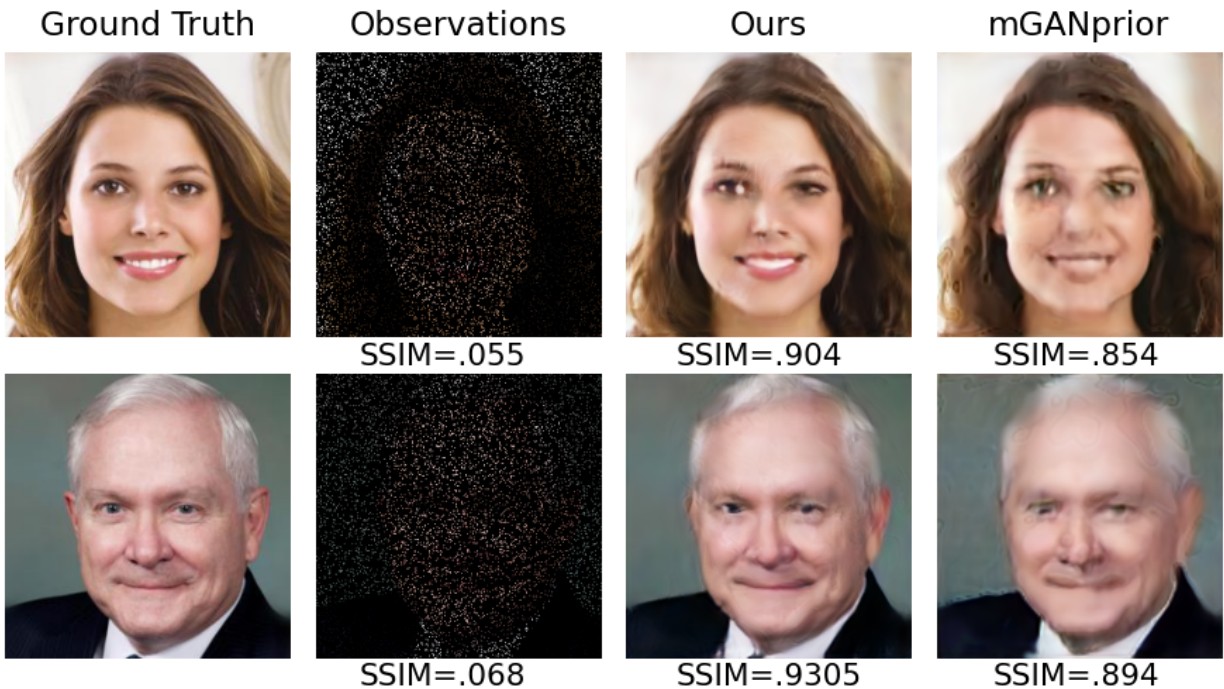

Figure 17: Qualitative comparison between our method mGANprior-RTIL and mGANprior for inpainting at 10% of observed pixels.

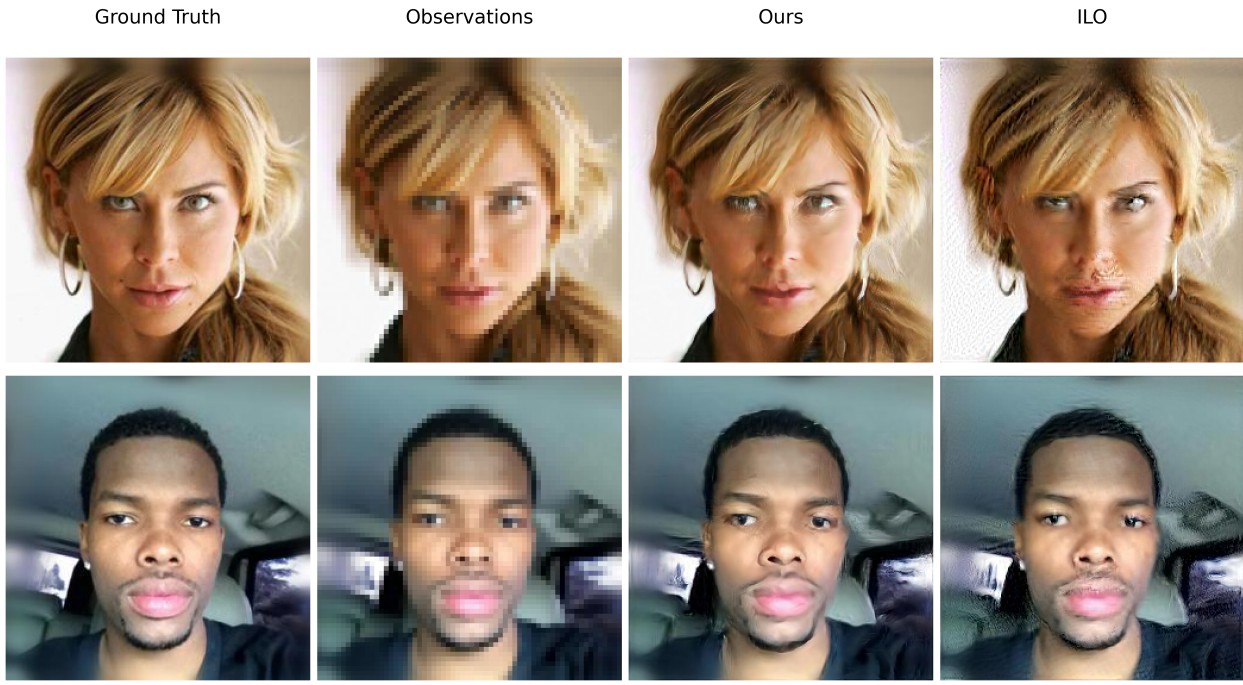

Figure 18: Comparison between ILO-RTIL to ILO for super-resolution LR 4x (Downsampling factor $\frac{1}{4}$)

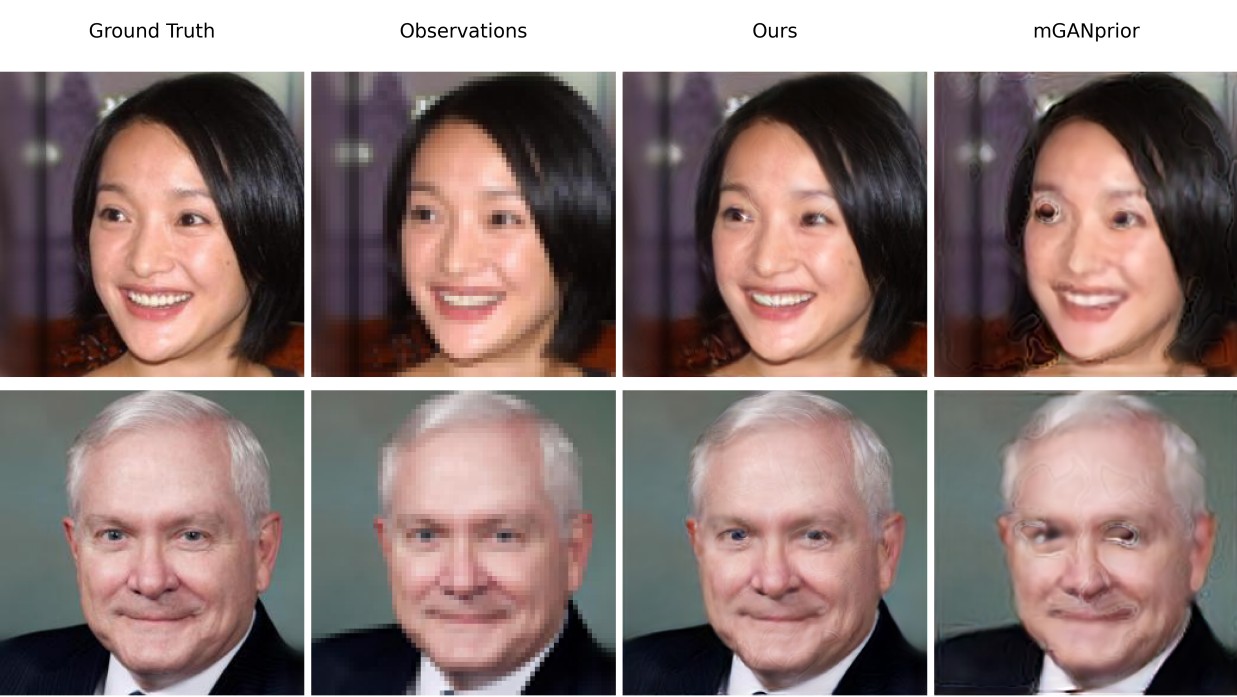

Figure 19: Comparison between mGANprior-RTIL to mGANprior for super-Resolution LR 4x (Downsampling factor $\frac{1}{4}$)

### A.6 Ablation

### A.6.1 ILO-Ablation

This section corresponds to Section 4 .3.1, Figure 6, the configuration for each optimization setting go as: $z_0 = \{2000\}$, $z_1 = \{2000, 2000\}$, $z_2 = \{2000, 1000, 2000\}$ , $z_3 = \{2000, 1000, 1000, 2000\}$, $z_4 = \{2000, 1000, 1000, 1000, 2000\}$ iterations per layer.

Figure 20 demonstrates how reconstruction performance can vary based on the number of iterations per intermediate layer while using ILO algorithm. The configuration for EXP 1-5 goes as following in order of number of iterations per intermeidate layer: EXP 1 = $\{100, 100, 100, 100, 100\}$, EXP 2 = $\{300, 300, 300, 300, 300\}$, EXP 3 = $\{1000, 1000, 1000, 1000, 1000\}$, EXP 4 = $\{2000, 1000, 1000, 1000, 2000\}$, EXP 5 = $\{2000, 2000, 2000, 2000, 2000\}$.

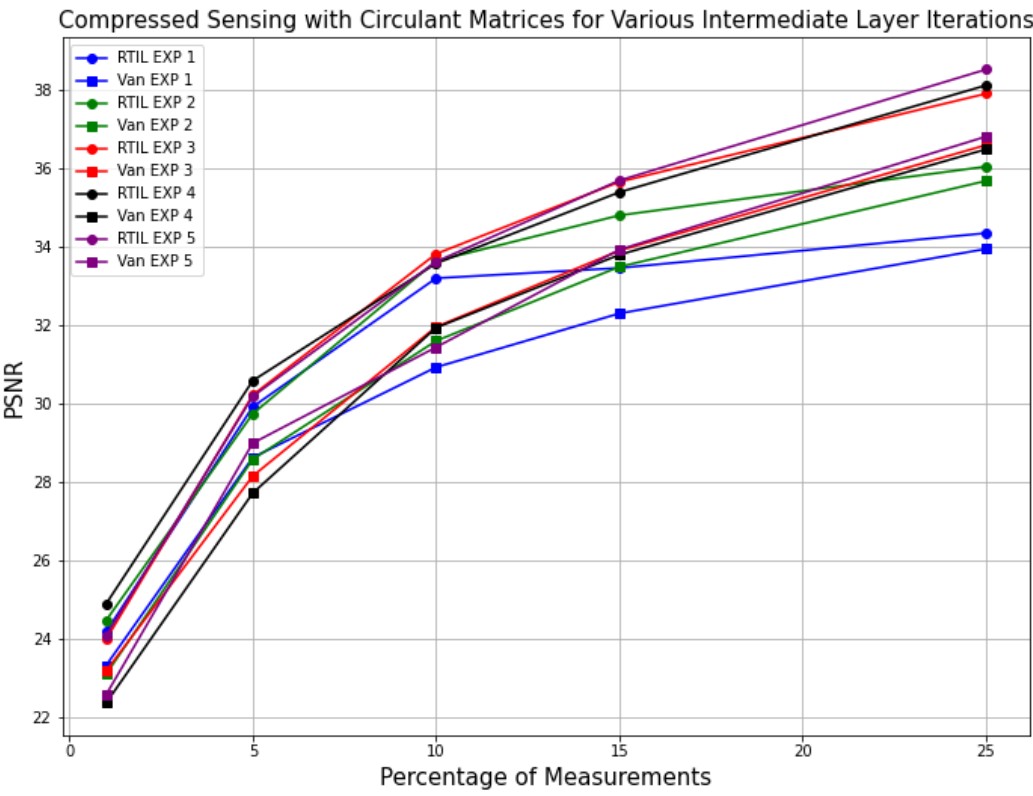

Figure 20: Compressed Sesning with Circulant Matrices reconstruction performance for various number of iteration per intermediate layer with ILO.

Figure 21 demonstrates how reconstruction performance can be affected based on the size of $\ell_1$ balls used in ILO algorithm 1. The number of iterations for each of the experiments $\{2000, 1000, 1000, 1000, 2000\}$ corresponding to each intermediate layer, which is The same used in the paper. Size of the $\ell_1$ balls were determined based on the author's recommendation in the appendix Daras et al. (2021). EXP 1-4 goes as following in order of number intermeidate layer and the corresponding radius of the size of search space for the $\ell_1$ ball of radius of noise, latent code, and projection of the previous solution: 1-$\{100, 200, 500, 1000, 2000\}$,$\{100, 200, 300, 800, 1000\}$,$\{100, 200, 300, 800, 1000\}$,2-$\{200, 1000, 100, 2000, 4000\}$,$\{200, 400, 800, 1600, 3200\}$,$\{200, 400, 800, 1600, 3200\}$,3-$\{1500, 1500, 2500, 4000, 5000\}$,$\{300, 500, 1000, 2000, 3000\}$,$\{300, 1200, 2000, 3000, 5000\}$,4-$\{500, 1500, 2500, 4000, 5000\}$,$\{500, 750, 1500, 2500, 4000\}$,$\{500, 750, 1500, 2500, 5000\}$.

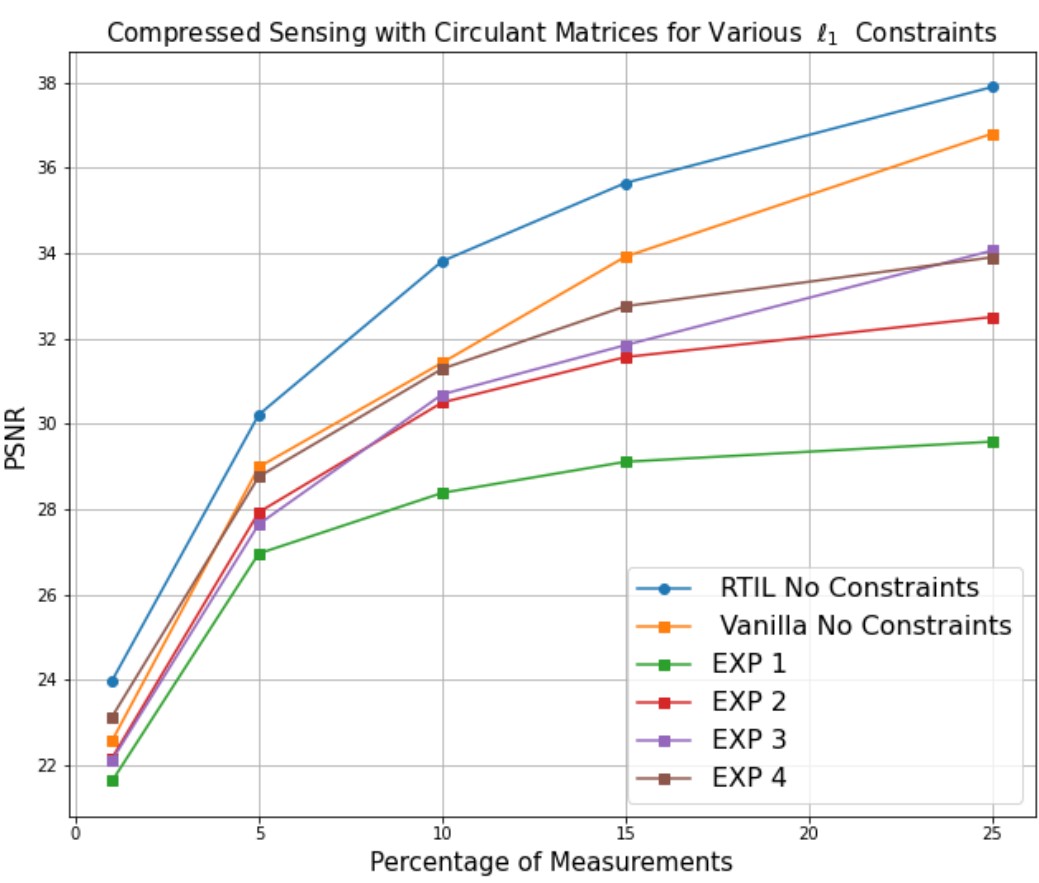

Figure 21: Compressed Sesning with Circulant Matrices reconstruction performance for various $\ell_1$ constraints.

### A.6.2 mGANprior-Ablation

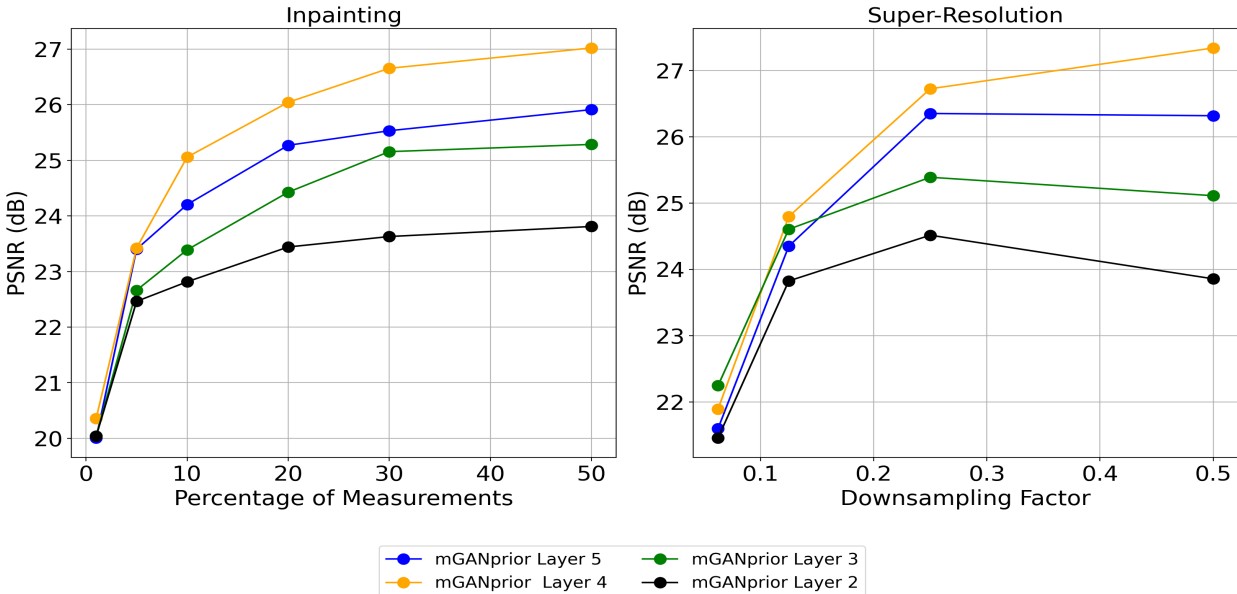

Figure 22: Effects on in-painting and super-resolution performance for various intermediate layer on validation set of 5 images.

