# OpenReview forum: "Regularized Training of Intermediate Layers for Generative Models for Inverse Problems"
_TMLR — Accepted by TMLR_

### Review · Reviewer_dJAR · 2022-11-12

**Summary Of Contributions:**

This paper propose to train the intermediate layers of GANs in order to do better GAN inversion, which can be further applied to inverse problems by serving as a generative prior. The method is based on existing GAN inversion methods such as Intermediate Layer Optimization and Multi-Code Prior. These methods optimize over intermediate layers by introducing an additional optimization variable, and  RTIL further trains the GAN model with GAN objective. In other words, it optimize the GAN parameters with sampling from the extra latent variables. Results show some improvements in GAN inversion (reconstruction error) and applications to inverse problems (super-resolution, in-painting, compress sensing).

**Audience:**

Yes

**Claims And Evidence:**

Yes

**Requested Changes:**

1. Clearly discuss the computational overhead issue.

2. Compare with generative models that solve inverse problem directly, and with more meaningful metric such as FID.

**Strengths And Weaknesses:**

Strengths:

The paper is clearly written, with nice figure illustration. The idea is a natural but meaningful extension over existing GAN inversion methods, and it shows some promising results on different tasks.

Weakness:

1. Not discussing the details of computational overhead. I expect that the inversion results will be better when we introduce the new training method, however, training a StyleGAN can be very expensive. In contrast, previous inversion methods can use pre-trained models. Therefore, it is important to discuss this trade-off. How much computational overhead is introduced?

2. Related to previous one. It is questionable that whether it worths to train a new model, given that the improvements are not very significant. In particular, multiple previous work train generative models directly to solve inverse problems (instead of serving as a prior) by modeling the conditional distribution. Since your method now requires training a new generative model, those method should be compared with. For example, super-resolution [1] and inpainting [2]. These are based on diffusion models, but there are many others based on GAN.

3. Evaluation metric. Work on using generative models to do super-resolution or inpainting need to report FID, rather than single-image metric like PSNR on a few images. It is insufficient to evaluate on a few images as that may result from cherry picking.

[1] Image Super-Resolution via Iterative Refinement https://arxiv.org/abs/2104.07636

[2] RePaint: Inpainting using Denoising Diffusion Probabilistic Models, https://arxiv.org/abs/2201.09865

---

> ### Author Response · Authors · 2022-12-10
> **Author Response**
>
> $\textbf{What is the computational overhead?}$ The computational overhead is stated in Figure 1, it is the cost of training. We initially tried training from a checkpoint instead of training from scratch, however, the deeper intermediate layers were unable to generate high-quality samples which resulted in the GAN collapsing. One could imagine future efforts how to stabilize this training process. Moreover, when comparing the training of the same model, i.e. stylgan2, there is no additional cost between vanilla and RTIL. Our method utilizes parameter sharing to train a family of generative models (Figure 3 and Figure 12) and our experiments validate this because they use the same number of parameters and achieve similar FID scores. Refer to the table given in RRQ response about FID scores.
>
> $\textbf{Comparing RTIL to other methods that solve inverse problems directly?}$ Our work deviates from the training generative model for a specific type of inverse problem (i.e. super-resolution/inpaitining) by training the model for a particular type of inversion algorithm. Therefore, our method is trained unsupervised of the forward operator and can be adapted to a wide range of different types of inverse problems. However, we do agree it is an interesting and valuable comparison between supervised and unsupervised methods, but it is outside the scope of this paper.
>
> $\textbf{Additonal metrics?}$ Please refer to the response given in UPvB about additional experiments/metrics.

---

### Review · Reviewer_UPvB · 2022-11-25

**Summary Of Contributions:**

This paper studies how to improve a specific class of signal inversion methods based on generative models. More specifically, it improves the inversion methods that reconstruct the original signal by optimizing the intermediate features of a generative model.

While generative models are trained to generate a target output $y$ from an input $x_0$, i.e. $y = g(x_0)$, where $g$ is a composition of multiple functions $g_i(*)$: $g(x_0) = g_t \circ g_{t-1} \circ ... \circ g_1 \circ g_0(x_0)$, these methods recover not only $x_{0}$ but also $x_{i}$ in order to recover the true $y$, i.e. from
\begin{equation}
arg min_{x_0} || y - g(x_0) ||
\end{equation}
to
\begin{equation}
arg min_{x_0, ..., x_{i}} || y - g(x_0, ..., x_{i}) ||.
\end{equation}

Here $x_{i}$ stands for the input of $g_i(\*)$. This can be achieved by relaxing $x_{i} = g_{i-1} (x_0)$, e.g. $x_{i} = g_{i-1} (x_0) + z_{i-1}$, where $z_{i-1}$ are additional variables in the signal reconstruction stage and are not used during training $g(*)$.

The main idea of the proposed method is, instead of training $g(x_0)$, we should train $g(x_0, ..., x_{i})$ directly for the inversion problem. Empirical results on two existing methods show that the proposed method outperforms the baselines using $g(x_0)$.

**Audience:**

Yes

**Broader Impact Concerns:**

No specific concern.

**Claims And Evidence:**

No

**Requested Changes:**

Given the limited technical contribution, I think a more detailed empirical evaluation is necessary for acceptance. In particular, the authors should:
* Evaluate the proposed method on more diverse tasks and data. Face images are not sufficient to demonstrate the usefulness of the proposed method.
* Evaluate on larger datasets with standard metrics to really verify the performance.
* Compare with state-of-the-art methods for the tasks used in evaluation.

Also, I think it is not sufficient to show that the proposed method improves the corresponding baseline. The authors should explain why this is important and the improvement really matters in practice.

**Strengths And Weaknesses:**

Strength
* the proposed approach is intuitive and is generic to an array of signal inversion / reconstruction methods
* Empirical results on multiple existing image inversion methods look promising

Weakness
* The technical contribution is limited, as the proposed method is a straightforward extension of existing methods.
* The evaluation is very limited from multiple aspects, including
  * Models: the experiments only consider two models, one for each inversion method
  * Data and task: the experiments are performed only on FFHQ and on a small number of related tasks
  * Test data size: it is very small and not comparable with related works in super resolution, etc.
  * Evaluation metrics: only PSNR is reported, while other standard metrics like LPIPS, FID, are not considered
  * Baselines: the proposed method is not compared with other existing methods for the inversion problem and methods for inpainting / super-resolution / compressed sensing

---

> ### Author Response · Authors · 2022-12-10
> **Author Response**
>
> $\textbf{The significance of paper} $ While RTIL indeed focuses on a particular class of inversion methods, multiple papers [Daras et al,Gu et al,Smedemark-Margulies et al] have introduced new methods of this class and have shown that they have had good performance for image reconstruction.    We show that our method improves both ILO and mGANprior, which are different algorithms within this class.  We believe our work points out to the community that post-hoc inversion algorithms can give rise to novel training algorithms.  We believe that this point is not widely appreciated and future applied work would benefit from the knowledge presented in our paper.
>
> $\textbf{ Adding more experiments}$ The test set has increased from $N=60$ to $N=100$ randomly sampled images for CELEBA-HQ and updated the metrics. We included LPIPS while keeping the original metrics reported in the paper PSNR and SSIM (Appendix). Please refer to the links to view the figures. Since the experiments are computationally expensive across the various different models and different type of inverse problems we were only to abe increase the test set by 40 images.
> PSNR and SSIM higher is better  LPIPS lower is better
> RTIL-ILO PSNR - https://i.postimg.cc/wj45cz1v/inv-ILO-error-bars-psnr.png
> RTIL-ILO SSIM - https://i.postimg.cc/BQ9JbPNZ/inv-ILO-error-bars-ssim.png
> RTIL-ILO LPIPS - https://i.postimg.cc/3xGppGQL/inv-ILO-error-bars-lpips.png
> RTIL-mGANprior PSNR - https://i.postimg.cc/CK3w8ZY4/inv-mgan-error-bars-psnr.png
> RTIL-mGANprior SSIM - https://i.postimg.cc/rFP3sBq1/inv-mgan-error-bars-ssim.png
> RTIL-mGANprior LPIPS - https://i.postimg.cc/QMGX99jJ/inv-mgan-error-bars-lpips.png
>
> $\textbf{Comparing to state of the art}$  Our goal is not to weigh in on the current state-of-the-art for any particular inverse problem, but instead, critique a current trend in the field. We demonstrate, if you are going to use post-hoc inversion methods, you should train the relevant generative model using RTIL instead of using an off-the-shelf model that was trained agnostic to the inversion
> algorithm.

---

### Review · Reviewer_RRRQ · 2022-11-28

**Summary Of Contributions:**

The paper proposes a novel way of solving inverse problems with GANs. The paper builds upon two well-established prior works, ILO and Multi-Code Prior. Both prior works attempt to increase the expressive power of the generator by optimizing intermediate layers. The authors of this work argue that when doing so, the intermediate layers need to be regularized during training. The propose a novel adversarial loss that feeds to the discriminator samples from both the vanilla model and the "expanded" model with intermediate latents coming from a simple distribution. This imposes constraints on the intermediate learned manifolds and experimentally yields performance benefits over the baseline methods. Theoretically, the authors analyze a toy setting under which they prove that their method is provably superior to ILO.

**Audience:**

Yes

**Broader Impact Concerns:**

I do not have any concerns on the ethical implications of this work.

**Claims And Evidence:**

Yes

**Requested Changes:**

* From Figure 1, it seems that first the network is trained with vanilla training and then finetuned. In the experimental section, it seems that the regularized models are trained from scratch (?). Could the authors please clarify? How is the performance impacted when the regularized models are trained from scratch vs. being finetuned from a pre-trained checkpoint?

* For the models trained from scratch, I would like to see unconditional FID scores. The objective here is not to achieve state-of-the-art, but to eliminate the possibility that better performance in inverse problems comes from having a better unconditional generator.

* The proposed regularization is to sample intermediate latents from a simple distribution during training. However, during inference, the optimization in the intermediate latent space is unconstrained (as far as I understand). It would be very interesting to see if the latents obtained by solving inverse problems are actually closer to being sampled from a Gaussian in the case of RTIL-ILO vs ILO.

* For the toy setting proof, how would things change if $W_0^*$ was also an optimization variable? It seems that a central argument for the proof relies on the fact that the RTIL-ILO has a unique solution (for this toy setting), while the vanilla method has infinite solutions that might introduce errors in certain directions. I would like the authors to clarify this point.

* From the plots of Figure 4, it seems that RTIL-ILO has a bigger advantage over ILO when the measurements are high. What is the intuition behind this observation? I would think that regularization is less required as more and more measurements are observed. Could the authors give numbers on what happens when even more measurements are available? Is ILO finally catching up to RTIL-ILO?

* Minor: there are some typos in Equations (6), (7).

**Strengths And Weaknesses:**

Strengths:

* The idea of the paper is novel and intuitive. Prior work has shown that i) extending the generator range and ii) regularizing the solutions are both important steps for solving inverse problems. This paper proposes a regularization during the training phase.
* The proposed approach maintains the flexibility of solving inverse problems with deep priors; the network is only trained once and then can be used to solve many inverse problems.
* The paper is clearly written.
* The comparison with the baselines is thorough. The authors compare in many tasks, including Compressed Sensing, Inpainting and Super-Resolution. The authors run a fair amount of ablation studies to show that the comparison with the baselines is fair.


Weaknesses:

* There is an "unfair" advantage of this work over ILO and multi-code prior, namely the authors of this work are allowed to change the generator during training. It seems reasonable that by doing so the authors are able to obtain better results for inverse problems.
* To use this approach, one needs to change the training of the model. That can be problematic for a couple of reasons. First, it is quite common that the weights of Foundation Models are released without their training data. In that case, the loss of Eq. (2) cannot be used. Second, training a model (or even finetuning it) is much more computationally expensive compared to solving an inverse problem.
* Since the networks used in the paper are trained from scratch, it would be good to see their unconditional performance, e.g. maybe report FID scores. Sometimes, improved performance in inverse problems comes purely from having access to better generators.

---

> ### Author Response · Authors · 2022-12-10
> **Author Response**
>
> $\textbf{Clarification of Figure 1}$ The objective of Figure 1 is to demonstrate the workflow of using the RTIL principle. Step 1 is to train a generative network with latent variable $z_0$, sampled from a latent distribution $pz_0$, and outputting signals $x$ from the target distribution. The second step is to explore various inversion algorithms,
> including some that optimize over intermediate layers by introducing an additional optimization variable $z_1$. Step 1 can be skipped if you have access to a pre-trained model online, and start at step 2. Third, if such an algorithm provides
> competitive performance for inversion, then use RTIL to devise a new GAN training algorithm. This can be
> achieved by introducing a new latent variable $z_1 \sim pz_{1}$ where $pz_1$
> is an appropriate probability distribution. This model is trained from scratch where the latent variables are the free variables of the optimization algorithm during inversion. Step 1 and Step 3 use the same model, but the model in 1 is trained agnostic to the inversion algorithm used downstream, whereas the model in 3 is not.
>
> $\textbf{FID Score for Unconditional Generation} $  The table below provides FID Scores for both RTIL-ILO and RTIL-mGANprior. RTIL-ILO was trained with a stylegan2 architecture and RTIL-mGANprior with a progressively growing GAN both on FFHQ dataset. The numbers reported below are on a randomly sampled subset of 50,000 images training set, the average over 5 runs.
>
> ### Table stylegan2 (ILO)
>
> | Model  |    FID |
> | ------------- |:-------------:|
> | $G^{van}$ | 16.45    |
> |   $G_{0}^{RTIL}$ | 16.99     |
> | $G_{1}^{RTIL}$      | 17.609    |
> | $G_{2}^{RTIL}$      | 17.87    |
> | $G_{3}^{RTIL}$      | 17.99    |
> | $G_{4}^{RTIL}$      | 20.013    |
>
>
> ### Table PG-Gan (mGANprior)
>
> | Model  |    FID |
> | ------------- |:-------------:|
> | $G^{van}$ | 38.16   |
> |   $G_{0}^{RTIL}$ | 40.65    |
> | $G_{1}^{RTIL}$      | 41.34    |
> | $G_{2}^{RTIL}$      | 42.19    |
>
> $\textbf{Explanation of Theory}$ The Theorem illustrates
> that post-hoc addition of a free variable within a generative
> model can yield worse reconstruction error than when using
> a generative model trained with the added variable. The
> point is demonstrated in the simplest case that we could
> find that illustrates the effect: the case of linear regression, a
> purely supervised setup, and a known first layer. In principle,
> we could weaken the assumptions, but that would likely shed
> no additional fundamental insight into the theorem that we
> establish as additional complexities would enter, such as if $W_{0}^{\star}$ is not know. For example, if $W_{0}^{\star}$ is not known, then there are infinitely many solutions and the problem would require some type of regularization to specify a solution.
> However, this would not change the overall story because in this case RTIL would exactly recover $W_{0}^{\star}$ and $W_{1}^{\star}$, while, similar to the current version of Lemma 5.1, the vanilla training would only achieve $W_1 W_0 = W_1^\star W_0^\star$. Thus again $W_{1}$ in the vanilla model will have errors in the orthogonal complement of the range of $W_{0}^\star$ which would leads to unavoidable errors in the compressed sensing problem.
> We do not claim that the theorem provides a particularly deep mathematical claim, but it does allow for a formal observation of why RTIL works, at least in a simplified case.
>
> $\textbf{Intuition on why the difference between their method and unregularized case decreases as the problem becomes harder?} $ In the case of ILO, initially, the optimization problem is solved at the input layer $z_0$ then is refined at the next layer $z_1$. Therefore, if there are sufficiently few measurements that can be approximated by $z_0$, then there are no subsequent measurements for $z_1$ to fit.

---

### Decision · Action_Editors · 2023-01-17

**Recommendation:** Accept as is

**Comment:**

The submission looks into the problem of model inversion via optimization of intermediate activation values in generative adversarial networks (GANs). It proposes to make GANs more suitable for this kind of model inversion through regularized training (a technique called Regularized Training of Intermediate Layers, or RTIL). RTIL works by incorporating the latent variables introduced by inversion techniques like ILO or mGANprior into the training procedure itself. The GAN loss term which operates on generated data is replaced with a convex combination of loss terms which operate on data produced by the unmodified generator and data produced by the generator augmented with the additional latent variables prescribed by the chosen inversion algorithm.

The approach is evaluated and compared against the baseline of no regularized training for two pairs of inversion algorithm and generator architecture (ILO and StyleGAN2, mGANprior and PGGAN) and for three tasks (compressed sensing, inpainting, and super-resolution). Ablations are run to measure the efficacy of the proposed approach in comparison to the baseline in terms of number of latent codes introduced. Finally, theoretical results are presented in a toy setting to justify the soundness of the proposed approach.

Reviewers note the presentation's clarity (dJAR, RRRQ), the proposed approach's broad applicability (UPvB) and its flexibility (RRRQ), and the promising (UPvB) and thorough (RRRQ) empirical results.

Concerns raised by reviewers include:

- Reviewer RRRQ raises questions on the RTIL workflow shown in Figure 1, which the authors address in their response.
- Reviewer RRRQ asks for FID metrics for unconditional generation in order to rule out the possibility that the improvements observed simply stem from a better generator. The authors provide those metrics in their response.
- Reviewer RRRQ notes that the approach is not applicable to foundation models for which the training data is unavailable. While this is true, this does not detract from the paper's main claims.
- Reviewer UPvB finds the submission's technical contribution somewhat limited. TMLR's acceptance criteria are a) claims supported by accurate, convincing and clear evidence and b) interest to at least some individuals in TMLR's audience, therefore this concern is beyond the scope of TMLR's criteria.
- Reviewer UPvB is concerned that the evaluation is too limited in terms of models, tasks, data regimes, metrics, and baselines. The authors respond by increasing the test set size for CELEBA-HQ and providing LPIPS metrics. This does not satisfy Reviewer UPvB, who remains concerned that the empirical evidence is insufficient.
- Reviewer dJAR asks for a discussion of the computational overhead associated with the proposed approach, which the authors provide in their response.
- Reviewer dJAR is concerned that the proposed approach is not compared against approaches which train generative models directly to solve the inverse problem. The authors respond that while interesting, such a comparison is beyond the scope of the claims made in the paper.

Ultimately, all reviewers indicate that the submission is of sufficient interest to at least some individuals in TMLR's audience, and most reviewers indicate that the submission's claims are supported by accurate, convincing, and clear evidence. While I agree with Reviewer UPvB that expanding the evaluation would strengthen the submission, I side with the majority opinion that the empirical evidence presented is sufficient to support the paper's main claims.

**Audience:**

Generative modelling and inverse problems are active research areas, and the submission is of interest to those communities, as evidenced by the reviewers' official recommendation and explicitly noted by Reviewer RRRQ in their recommendation.

**Claims And Evidence:**

The main claim made by the paper is that if one is aware that a GAN will be used for inverse problems, then incorporating that knowledge as a regularizer to train the model improves the performance of model inversion algorithms. Most reviewers indicate that this claim is supported by accurate, convincing, and clear evidence.

---

> ### Author Response · Authors · 2023-02-10
> **Acceptance**
>
> Hello All,
>
> I would like to thank the AE and each reviewer for their positive comments and constructive suggestions for our work. We will be posting the camera-ready version soon.